# *SCN4B* acts as a metastasis-suppressor gene preventing hyperactivation of cell migration in breast cancer

Emeline Bon[1], Virginie Driffort[1], Frédéric Gradek[1], Carlos Martinez-Caceres[2], Monique Anchelin[3], Pablo Pelegrin[2], Maria-Luisa Cayuela[3], Séverine Marionneau-Lambot[4], Thibauld Oullier[4], Roseline Guibon[1,5], Gaëlle Fromont[1,5], Jorge L. Gutierrez-Pajares[1], Isabelle Domingo[1], Eric Piver[5,6], Alain Moreau[6], Julien Burlaud-Gaillard[7], Philippe G. Frank[1], Stéphan Chevalier[1,8,*], Pierre Besson[1,8,*] & Sébastien Roger[1,9,10,*]

The development of metastases largely relies on the capacity of cancer cells to invade extracellular matrices (ECM) using two invasion modes termed 'mesenchymal' and 'amoeboid', with possible transitions between these modes. Here we show that the *SCN4B* gene, encoding for the β4 protein, initially characterized as an auxiliary subunit of voltage-gated sodium channels (Na$_V$) in excitable tissues, is expressed in normal epithelial cells and that reduced β4 protein levels in breast cancer biopsies correlate with high-grade primary and metastatic tumours. In cancer cells, reducing β4 expression increases RhoA activity, potentiates cell migration and invasiveness, primary tumour growth and metastatic spreading, by promoting the acquisition of an amoeboid–mesenchymal hybrid phenotype. This hyperactivated migration is independent of Na$_V$ and is prevented by overexpression of the intracellular C-terminus of β4. Conversely, *SCN4B* overexpression reduces cancer cell invasiveness and tumour progression, indicating that *SCN4B*/β4 represents a metastasis-suppressor gene.

[1] Inserm UMR1069, Nutrition, Croissance et Cancer, Université François-Rabelais de Tours, 10 Boulevard Tonnellé, 37032 Tours, France. [2] Inflammation and Experimental Surgery Unit, CIBERehd, Murcia's BioHealth Research Institute IMIB-Arrixaca, Clinical University Hospital Virgen de la Arrixaca, E-30120 Murcia, Spain. [3] Telomerase, Cancer and Aging Group, Hospital Virgen de la Arrixaca, E-30120 Murcia, Spain. [4] Cancéropôle du Grand Ouest, Plateforme In Vivo, 44000 Nantes, France. [5] CHRU de Tours, 2 Boulevard Tonnellé, 37000 Tours, France. [6] Inserm, U966, Université François-Rabelais de Tours, 10 Boulevard Tonnellé, 37032 Tours, France. [7] Laboratoire de Biologie Cellulaire-Microscopie Electronique, Faculté de Médecine, Université François-Rabelais de Tours, 2 Boulevard Tonnellé, 37000 Tours, France. [8] UFR Sciences Pharmaceutiques, Université François-Rabelais de Tours, 31 Avenue Monge, 37200 Tours, France. [9] UFR Sciences et Techniques, Département de Physiologie Animale, Université François-Rabelais de Tours, Parc de Grandmont, 37200 Tours, France. [10] Institut Universitaire de France, 1, Rue Descartes, 75231 Paris Cedex 05, France. * These authors contributed equally to this work. Correspondence and requests for materials should be addressed to S.R. (email: sebastien.roger@univ-tours.fr).

The acquisitions of extensive invasion potencies by cancer cells are key components in the metastatic cascade, hence in patients survival[1,2]. During the last decade, important knowledge in various processes of cancer cell migration and invasiveness has emerged[3]. In the 'mesenchymal mode', engaged cells harbour an elongated fibroblast-like morphology, with a rear-to-front lamellopodial cell polarity, and generate a path in the extracellular matrix (ECM) through proteolytic remodelling. This is performed by invadosomal structures, which are F-actin-rich organelles, protrusive into the ECM and responsible for its proteolysis through the recruitment of both membrane-associated and extracellularly released soluble proteases[4,5]. In the other invasive mode, called 'amoeboid', cancer cells show no obvious polarity but a rounded morphology, and display high potentials for migration and invasiveness[6]. In this case, strong actomyosin contractions propel the cell, which deforms and squeezes inside small gaps of the ECM, with no need to degrade it. While different cancer cell types may preferentially engage into the mesenchymal mode or the amoeboid one, the most aggressive cancer cells show high plasticity and are able to switch from one phenotype to the other[6]. Such transitions, orchestrated by RhoGTPases family members[7–9], offer selective advantages and compensatory mechanisms to migrating cancer cells, presumably abrogating the efficacy of anticancer treatments[10]. Indeed, attempts to reduce cancer cell invasiveness and metastatic dissemination by targeting proteolytic activity and ECM remodelling have largely failed because of the adaptive compensatory mechanism sustaining protease-independent processes[11].

Voltage-gated sodium channels ($Na_V$) are composed of one large pore-forming principal subunit (nine genes encoding nine proteins, $Na_V1.1–1.9$)[12,13] and one or two smaller transmembrane subunits considered as auxiliary (four genes SCN1B to SCN4B, generating four subunits, β1 to β4)[14]. The activity of $Na_V$ gives rise to $Na^+$ currents ($I_{Na}$) generating action potentials in excitable cells such as neurons, skeletal and cardiac muscle cells. As a result, these proteins have been considered hallmarks of excitable cells. However, multiple studies have recently demonstrated their expression in non-excitable cells, in which they regulate cellular functions such as migration, differentiation, endosome acidification, phagocytosis and podosome formation[15]. In addition, $Na_V$ are abnormally expressed in carcinoma cells and tumour biopsies, and their activity is associated with aggressive features and cancer progression[16–18]. Expression of the $Na_V1.5$ isoform in breast tumours is correlated with metastases development and patients' death[19,20]. In highly aggressive human breast cancer cells, the activity of pore-forming $Na_V1.5$ is not associated with cell excitability but with ECM degradation and cancer cell invasiveness[21], hence favouring metastases development[22,23]. $Na_V1.5$-dependent invasiveness is mediated through allosteric modulation of the $Na^+$–$H^+$ exchanger NHE1, subsequent acidification of the pericellular microenvironment and activation of extracellular acidic cysteine cathepsins[24–26]. Furthermore, $Na_V1.5$ sustains Src kinase activity, polymerization of actin and acquisition by cells of a spindle-shaped elongated morphology[26]. Altogether, these results indicate a critical role for $Na_V1.5$ in 'mesenchymal invasion'. The participation of non-pore-forming SCNxB/β subunits in oncogenic processes was not studied as extensively, with the exception of the β1 subunit[27], and their roles during metastatic progression remain largely unknown.

In this study, we show that the SCN4B/β4 subunit is expressed in normal epithelial cells and tissues, but is strongly down-regulated in aggressive cancer cells and tumours. The loss of SCN4B/β4 enhances cancer cell migration and metastases

formation through a RhoA-dependent signalling pathway, independently of pore-forming $Na_V$ subunits.

## Results

**Association with poor prognosis.** Expression of the β4 protein, encoded by the SCN4B gene[28], has mostly been studied in excitable cells in which mutations have been linked to sodium channelopathies[29,30]. Initial immunohistochemical analyses performed in normal and cancer breast tissues indicated that the SCN4B/β4 protein was specifically expressed in epithelial cells from normal mammary acini, but was significantly downregulated in cancer cells (Fig. 1a,b). Levels of β4 expression were analysed by immunohistochemistry on tissue microarrays containing normal breast, hyperplasic, dysplasia, breast cancer and lymph node metastases (LNM) samples. Again, the expression of SCN4B/β4 was strong in epithelial cells from normal non-excitable mammary tissues (Fig. 2a,b), as well as in breast hyperplasia (Supplementary Fig. 1). Levels of SCN4B/β4 expression in mammary hyperplasia and dysplasia were similar to those recorded in normal mammary tissues, but were remarkably reduced in biopsies of mammary carcinomas. The most important reductions in SCN4B/β4 expression were observed when comparing in situ grade I to invasive grade II breast

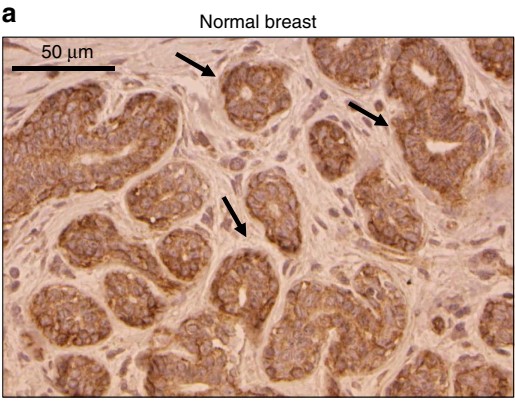

**a** Normal breast
50 μm

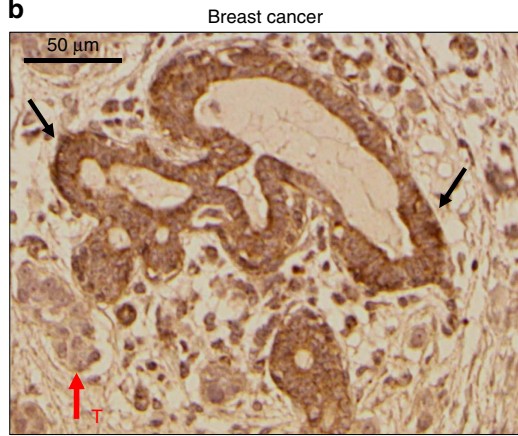

**b** Breast cancer
50 μm

**Figure 1 | SCN4B/β4 protein is expressed in normal epithelial cells of human breast tissues and is downregulated in cancer cells.**
(**a,b**) β4 protein (expression of the SCN4B gene) was analysed by immunohistochemistry on human breast tissue samples. (**a**) The expression of β4 protein was strong in epithelial cells of mammary acini (some examples are indicated by the black arrows), and not in non-epithelial cells of normal breast tissues. (**b**) In breast cancer tissue, the expression of β4 protein was strong in normal epithelial cells of mammary acini (black arrows), but significantly reduced in cancer cells (tumour area indicated by the red arrow, 'T'). Scale bars, 50 μm.

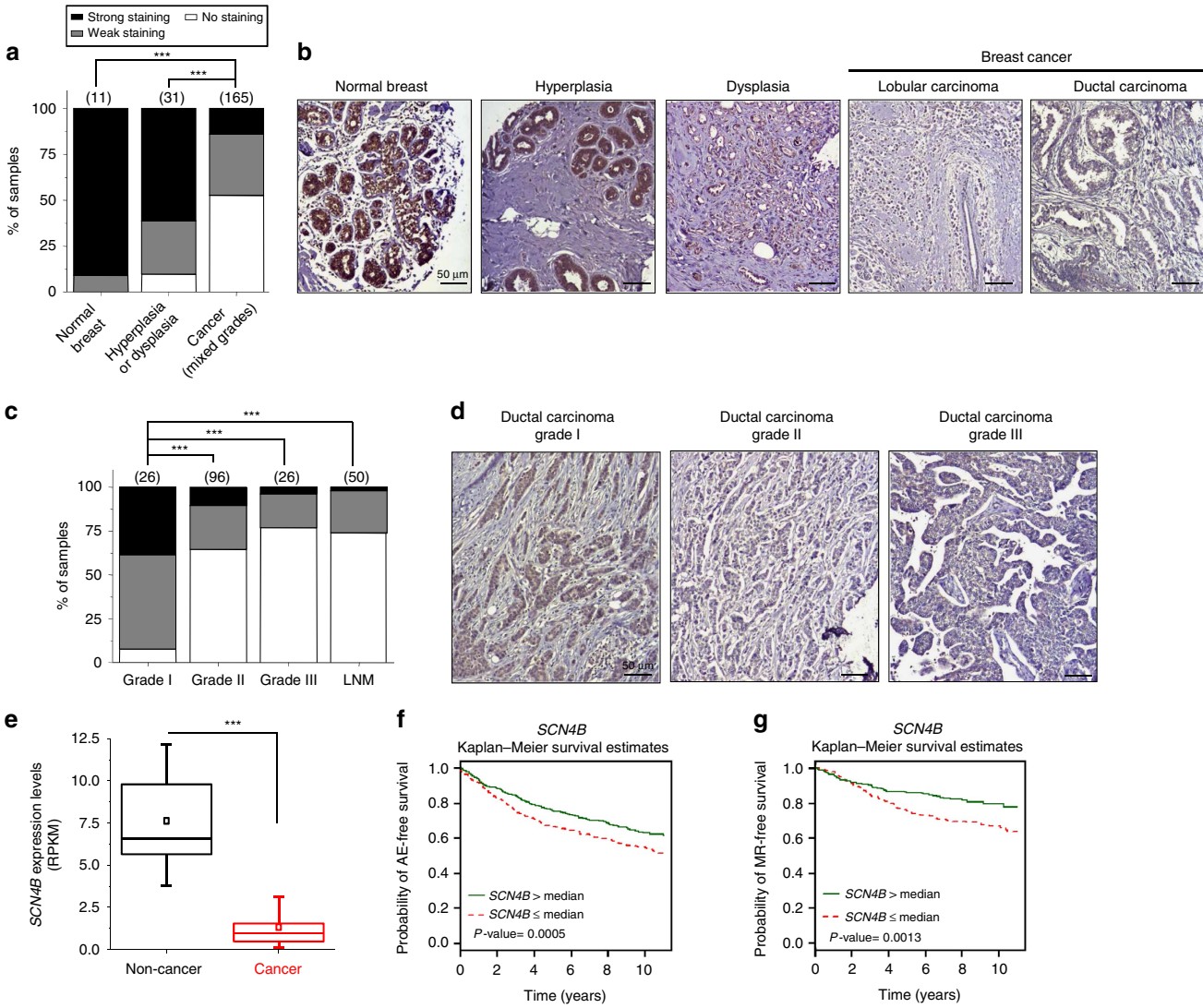

**Figure 2 | *SCN4B* down regulation in human breast cancer tissues associates with poor prognosis.** (**a–d**) *SCN4B*/β4 protein expression was analysed by immunohistochemistry on breast tissue microarrays. Samples were stratified in 'no staining', 'weak staining' or 'strong staining' groups. (**a**) Proportion of samples showing no (white), weak (gray) or strong (black) β4 staining in normal breast, compared with mammary hyperplasia/dysplasia and cancer (mixed grades) samples. The number of samples per condition is indicated in brackets. *SCN4B*/β4 protein staining was stronger in normal compared with cancer samples ($\chi^2$, $P < 0.001$), and in hyperplasia/dysplasia compared with cancer samples ($\chi^2$, $P < 0.001$). (**b**) *SCN4B*/β4 staining from indicated samples. Scale bars, 50 µm. (**c**) Proportion of samples showing no, weak or strong *SCN4B*/β4 staining in cancer samples, from grade I to III, and in lymph node metastases (LNM) samples. The number of samples per condition is indicated in brackets. β4 staining was stronger in grade I cancer samples compared with more advanced cancer samples (grades II, III and LNM) ($\chi^2$, $P < 0.001$). (**d**) Representative pictures of β4 staining from grade I, II and III ductal carcinoma samples. Scale bars, 50 µm. (**e**) Expression of the *SCN4B* gene in non-cancer ($n = 29$) and in invasive breast carcinoma tissues ($n = 145$) was analysed from The Cancer Genome Atlas (TCGA). RNA level is expressed as reads per kilobase per million (RPKM). Box plots indicate the first quartile, the median and the third quartile; whiskers indicate minimum and maximum values; squares show the means. *SCN4B* gene was significantly downregulated in cancer tissues (Mann–Whitney rank sum test, MW, $P < 0.001$). (**f,g**) Prognostic analyses of gene expression in breast cancers, performed using the *Breast Cancer Gene-Expression Miner*. (**f**) Kaplan–Meier Any Event (AE)-free survival analyses, performed on data pooled from cohorts for the expression of *SCN4B* gene ($n = 1,024$ patients). AE is defined as being metastatic relapse (MR) or patient death. A weak expression of *SCN4B* gene ($\leq$ median of the pooled cohorts) was associated with a decrease in the AE-free survival ($P = 0.0005$). (**g**) Kaplan–Meier MR-free survival analyses were performed for the expression of *SCN4B* ($n = 661$ patients). A weak expression of *SCN4B* ($\leq$ median of the pooled cohorts) was associated with a decrease in the MR-free survival ($P = 0.0013$). Cox results are displayed on the graph.

tumours. *SCN4B*/β4 protein was weakly expressed or totally absent in most grade II and III primary tumours studied, as well as in LNM (Fig. 2c,d). Lower levels of the *SCN4B* gene transcript in breast cancer tissues, compared with non-cancer tissues, were also found to be highly significant in RNA sequence expression analysed from The Cancer Genome Atlas Network (Fig. 2e) and to be associated with an increased risk of metastatic relapse or death in breast cancer patients (Fig. 2f,g). Levels of expression of

the other *SCNxB* genes were also assessed. Expressions of *SCN1B*, *SCN2B* and *SCN3B* were lower in cancer compared with non-cancer tissues (Supplementary Fig. 2a,c,e). However, there was no association between the levels of expression of these genes and the risk of metastatic relapse (Supplementary Fig. 2b,d and f). Importantly, *SCN1B* and *SCN4B* genes appeared to be the two most highly expressed *SCNxB* genes in non-cancer tissues (Supplementary Fig. 2g). Furthermore, *SCN4B* seemed to be the

most significantly downregulated *SCNxB* gene in breast cancer tissues compared with non-cancer tissues (Supplementary Fig. 2h). Similarly, the analysis of data from two published studies[31,32] showed that *SCN4B* expression levels were downregulated in lung cancer compared with normal lung tissues (Supplementary Fig. 3a,b) and our immuno-histochemical analyses in lung cancer tissue microarrays also identified a tendency for decreased protein expression in high-grade primary lung tumours and metastases (Supplementary Fig. 3c,d). *SCN4B* expression was also down-regulated in prostate, colon and rectal cancers compared with normal tissues (Supplementary Fig. 3e,g). This suggests that *SCN4B*/β4 is normally expressed in normal epithelial cells (also confirmed by immunohistochemical analyses of normal oesophagus (Supplementary Fig. 4) and that its loss of expression is associated with a gain in invasive properties by cancer cells and an aggressive progression of epithelial tumours. Correlatively, we found that *SCN4B* expression was higher in non-cancer epithelial mammary MCF-10A compared with several breast cancer cell lines such as MCF-7, MDA-MB-468, MDA-MB-435s and MDA-MB-231 (Fig. 3a,b). Particularly, the expression level of *SCN4B* was low in the highly invasive and metastatic MDA-MB-231 breast cancer cells, known to express functional Na$_V$1.5 (ref. 22). *SCN4B* mRNA (Fig. 3c) and protein (Fig. 3d) were expressed in MDA-MB-231 cells genetically modified with the luciferase gene (MDA-MB-231-Luc cells). *SCN1B*/β1 and *SCN2B*/β2 were also expressed, but not *SCN3B*/β3. The lack of expression of *SCN3B*/β3 was further demonstrated in conventional PCR using two different couples of primers (Supplementary Fig. 5a), thus confirming previously published results obtained with wild-type cells[24,33]. Transient silencing of *SCNxB* genes using specific small-interfering RNA (siRNA: si*SCN1B*, si*SCN2B* or si*SCN4B*) was responsible for significant (65–80%) decreases in protein levels 48 h after transfection, as compared with a control null-target siRNA (siCTL) (Fig. 3e). The downregulation of one of the *SCNxB* gene had no effect on the mRNA expression of the others, suggesting the absence of compensation between *SCNxB*/β subunits in these conditions (Supplementary Fig. 5b). While reduced expression of either *SCN1B*/β1 or *SCN2B*/β2 decreased cancer cell invasiveness through Matrigel-coated invasion chambers by 42.8 ± 6.6% ($n = 8$) and 51.7 ± 0.3% ($n = 8$) (mean ± s.e.m. (number of independent experiments)), respectively, inhibition of *SCN4B* expression enhanced invasiveness by 62.4 ± 12.2% ($n = 8$) (Fig. 3f,g). We also analysed the consequences of knocking-down the expression of *SCN4B* on MDA-MB-231-Luc invasiveness *in vivo* using the zebrafish model of micrometastasis[34,35]. Sixty-one per cent of zebrafish embryos injected with siCTL cells had their organs colonized. This number was increased to approximately 87% of the embryos presenting micrometastases when injected with si*SCN4B* cells, resulting in an increase in the zebrafish colonization index by 1.41 ± 0.08 fold ($n = 3$) (Fig. 3h,i). These quantifications were performed 72 h after siRNA transfection, a time for which *SCN4B* downregulation was still efficient (Supplementary Fig. 6a).

**Independence of pore-forming Na$_V$ subunits**. Breast cancer cell invasiveness has been demonstrated to be strongly regulated by the activity of the pore-forming Na$_V$1.5 (refs 19,20,24). Therefore, we initially hypothesized that loss of *SCN4B* expression would increase Na$_V$1.5 activity in highly aggressive cancer cells. To test this hypothesis, we generated MDA-MB-231-Luc-derived cell lines stably expressing a null-target small hairpin RNA (shCTL cells), or expressing shRNAs targeting either the expression of *SCN5A* gene encoding for Na$_V$1.5 (sh*SCN5A* cells, in which the

expression of the *SCN4B* gene is not changed) as previously described[22] or targeting *SCN4B* transcripts (sh*SCN4B* cells). The use of sh*SCN4B* resulted in 81.1 ± 0.2% ($n = 37$) decrease in mRNA levels (Supplementary Fig. 6b) with no effect on the expression of the other *SCNxB* subunits (Supplementary Fig. 6d,e). The three cell lines generated displayed identical viability and growth properties (Supplementary Fig. 6c). In shCTL cells, $I_{Na}$ is known to be generated by the sole Na$_V$1.5 isoform which, while commonly referred to as tetrodotoxin (TTX)-resistant, can almost completely be inhibited by 30 μM TTX[22,36]. Similarly to already reported results[24,37], the inhibition of Na$_V$1.5 sodium currents in shCTL cells using 30 μM TTX reduced invasiveness by 45.3 ± 4.3% ($n = 8$), similar to the reduction in invasiveness (43.4 ± 5.1%, $n = 6$) in sh*SCN5A* cells, not expressing Na$_V$1.5 and which were no longer sensitive to the addition of TTX (Fig. 4a). Knocking-down the expression of the *SCN4B* gene with different interfering RNA sequences resulted in similar potentiations of aggressiveness (sh*SCN4B*, Fig. 4a and si*SCN4B*; Fig. 3e,f). Sh*SCN4B* cell invasiveness was 281.8 ± 16.2% ($n = 16$) higher than that of shCTL cells. The treatment of sh*SCN4B* cells with 30 μM TTX, a concentration that inhibits all Na$_V$ channels, with the exception of the very TTX-resistant Na$_V$1.8, significantly reduce their invasiveness (Fig. 4a). To assess a possible independence from Na$_V$1.5 in the increased invasiveness mediated by the loss of *SCN4B* expression, we silenced *SCN5A* expression in sh*SCN4B* cells. This significantly reduced cancer cell invasiveness, which was still 184.3 ± 27.0% ($n = 12$) of shCTL cells (Fig. 4b). We then transiently inhibited the expression of *SCN4B* using si*SCN4B* siRNA in sh*SCN5A* cells, which no longer express Na$_V$1.5. These cells were also treated or not with (i) 3 μM TTX which inhibits all TTX-sensitive channels (Na$_V$1.1–1.4 and Na$_V$1.6–1.7) or (ii) 30 μM TTX to also inhibit Na$_V$1.5 and Na$_V$1.9 TTX-resistant channels, or (iii) with 30 nM of the specific Na$_V$1.8 inhibitor A803467[38]. As shown in Fig. 4c, silencing *SCN4B* expression with si*SCN4B* in sh*SCN5A* cells increased invasiveness ( + 251.7 ± 20.8%, $n = 6$) similarly to what was observed with sh*SCN4B* cells expressing Na$_V$1.5 (Fig. 4a). Furthermore, neither TTX nor A803467 reduced invasiveness of sh*SCN5A* cells transfected with siCTL or si*SCN4B* (Fig. 4c). Altogether, these data clearly demonstrate that the increased invasiveness of *SCN4B* gene-suppressed cells was not a consequence of the upregulation of Na$_V$1.5 or of any other Na$_V$. We also assessed the effect of knocking-down the expression of *SCN4B*, using siRNA, on the invasive properties of cancer cell lines, such as the non-small-cell human lung cancer H460 or the human prostate cancer PC3 cell lines, known to express functional Na$_V$ (Na$_V^+$) contributing to mesenchymal invasion. In addition, we used cancer cell lines that do not express Na$_V$ (Na$_V^-$) such as the less invasive human breast MDA-MB-468 and the non-small-cell lung A549 cancer cell lines[21,39,40]. In all cancer cell lines tested, showing similar levels of *SCN4B* expression (Supplementary Fig. 5d), transfection of si*SCN4B* significantly increased the invasiveness, from 23.7 ± 8.3% ($n = 3$) increase for PC3 cells up to 60.9 ± 9.5% ($n = 12$) increase in A549 cells (Fig. 4d), confirming that *SCN4B*/β4 loss potentiates invasiveness independently of the pore-forming Na$_V$ subunit.

Because Na$_V$1.5 has been reported as an important regulator of mesenchymal invasion in breast cancer cells through the potentiation of NHE1-dependent H$^+$ efflux and ECM degradation[22,25,26], we further investigated its regulation in sh*SCN4B* cells. Patch-clamp recordings revealed, contrary to our initial assumptions, that the loss of *SCN4B* expression significantly decreased the maximal peak $I_{Na}$, as observed in the $I_{Na}$–voltage relationship (Fig. 5a), suggesting a reduction of channel density at the plasma membrane. This could be due to the reduction of *SCN5A* expression levels as observed by

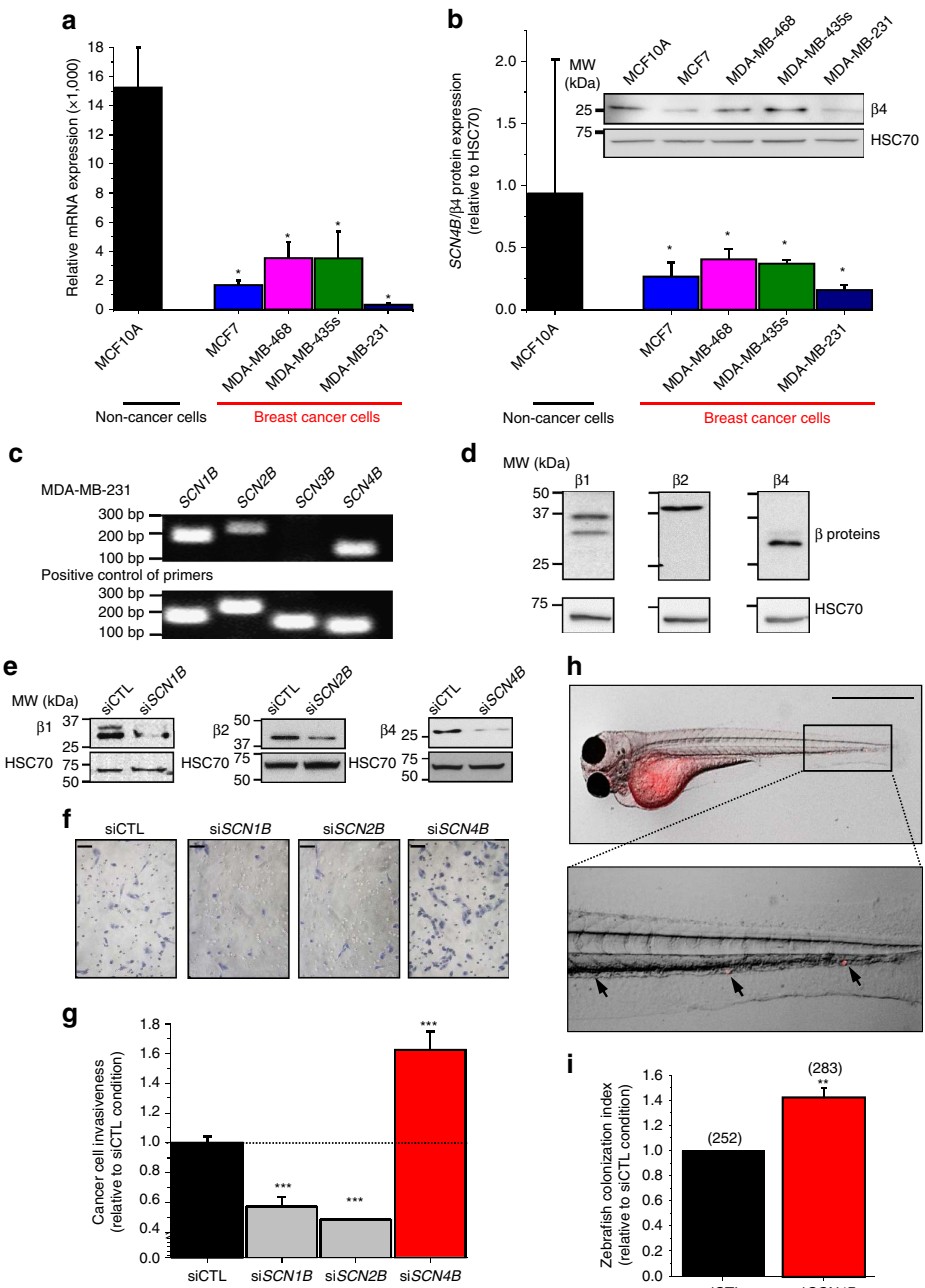

**Figure 3 | Expression of the *SCN4B*/β4gene in human breast cancer cell lines and contribution to cancer cell invasiveness.** (**a**) The expression of *SCN4B* gene was studied by RT–qPCR in human mammary epithelial non-cancer MCF-10A and cancer MCF7, MDA-MB-468, MDA-MB-435s and MDA-MB-231 cell lines. Results are expressed as relative to that of HPRT-1 gene ($n = 7$–12) and presented as mean values ± s.e.m. *, significantly different from MCF-10A at $P < 0.05$ (MW). (**b**) The expression of *SCN4B*/β4 protein was studied by densitometric analysis of western blot experiments in same cells as in **a**. Results are given as the amount of *SCN4B*/β4 protein relative to that of HSC70 ($n = 5$) and presented as mean values ± s.e.m. *, significantly different from MCF-10A at $P < 0.05$ (MW). The image on top shows a representative western blotting experiment. (**c**) The expression of *SCNxB* genes was analysed in MDA-MB-231-Luc cells by reverse transcription–PCR. Plasmids encoding human *SCNxB* genes were used as positive controls for PCR primers. (**d**) Representative western blotting experiments showing protein expression for β1 (*SCN1B*), β2 (*SCN2B*) and β4 (*SCN4B*) in MDA-MB-231-Luc cells. (**e**) Cells were transfected with scrambled siRNA (siCTL) or with siRNA directed against the expression of the *SCN1B* gene (si*SCN1B*), the *SCN2B* gene (si*SCN2B*) or the *SCN4B* gene (si*SCN4B*). The efficacy of siRNA transfection was assessed by western blotting experiments 48 h after transfection. HSC70 was used as a control for sample loading. (**f**) Representative images of fixed and haematoxylin-stained MDA-MB-231-Luc cells on invasion inserts. Cancer cells were transfected with scrambled siCTL or with specific siRNA. Scale bars, 50 μm. (**g**) Summary of cancer cell invasiveness results ($n = 8$) for MDA-MB-231-Luc cells transfected with siCTL or si*SCNxB*. Results were expressed relative to siCTL and presented as mean values ± s.e.m. ***, statistically different from siCTL at $P < 0.001$ (Student's *t*-test). (**h**) Representative image of a zebrafish embryo injected in the yolk sac with MDA-MB-231-Luc cells stained with CM-Dil and showing sites of colonization. Scale bars, 500 μm. Below is a magnification of the highlighted region containing human cancer cells (see arrows) colonizing organs of the embryo. (**i**) Zebrafish colonization index of siCTL or si*SCN4B* cells. Numbers in brackets indicate the number of embryos examined for each condition, from three different experiments. Results are presented as mean values ± s.e.m. **, statistically different from siCTL at $P < 0.01$ (Student's *t*-test).

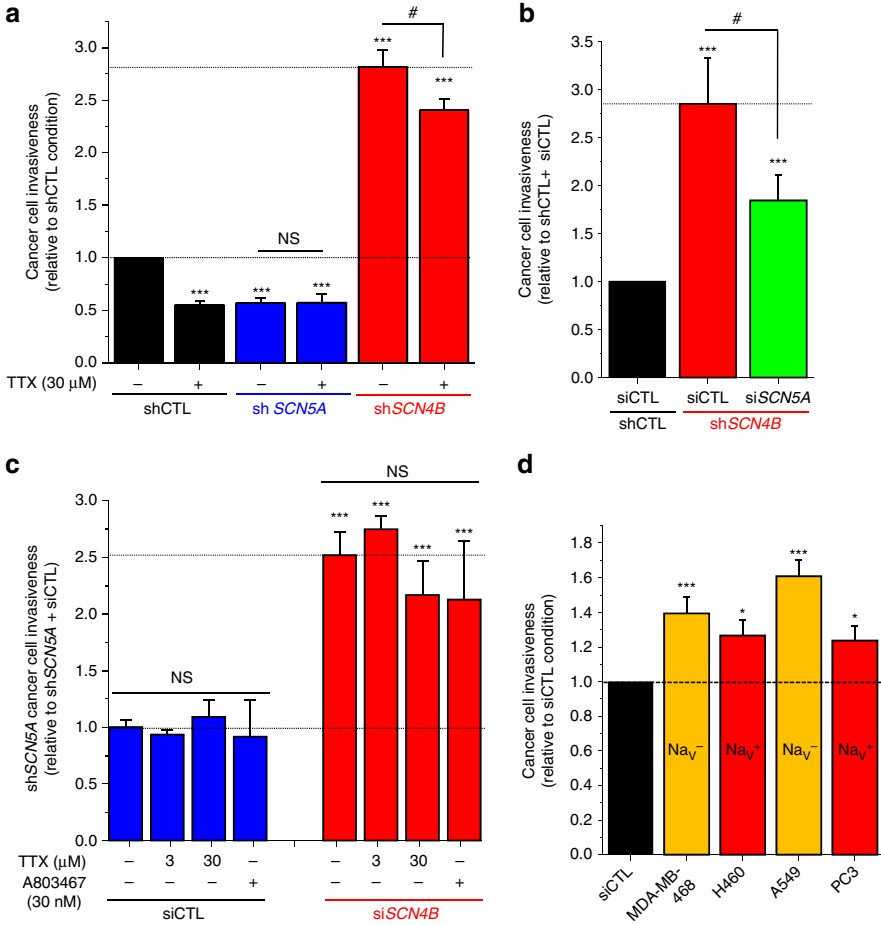

**Figure 4 | Loss of *SCN4B*/β4 expression promotes human cancer cell invasiveness independently of the pore-forming Na$_V$ subunit.** (**a**) Cancer cell invasiveness was assessed, using Matrigel-invasion chambers, from MDA-MB-231-Luc cells stably transfected with null-target shRNA (shCTL), *SCN5A*-targeting shRNA (sh*SCN5A*) or *SCN4B*-targeting shRNA (sh*SCN4B*), in the absence ( − ) or presence ( + ) of 30 μM TTX. The results from 8 to 16 independent experiments were expressed relative to control cells transfected with shCTL in the absence of TTX. ***, statistically different from shCTL at $P < 0.001$ and #, statistically different from sh*SCN4B* in the absence of TTX at $P < 0.05$. (**b**) Cancer cell invasiveness was likewise assessed in shCTL or sh*SCN4B* cells, transiently transfected with null-target siRNA (siCTL) or *SCN5A*-targeting siRNA (si*SCN5A*). The results from 12 independent experiments were expressed relative to shCTL cells transfected with siCTL. ***, statistically different from the shCTL/siCTL condition at $P < 0.001$ and #, statistically different from sh*SCN4B*/siCTL at $P < 0.05$. (**c**) Cancer cell invasiveness was assessed in MDA-MB-231-Luc cells stably expressing the *SCN5A*-targeting shRNA (sh*SCN5A*), not expressing the Na$_V$1.5 protein, and transiently transfected with null-target siRNA (siCTL) or *SCN4B*-targeting siRNA (si*SCN4B*). This effect was assessed in the absence ( − ) or presence ( + ) of two TTX concentrations (3 or 30 μM), or 30 nM of the Na$_V$1.8 inhibitor A803467. The results from six independent experiments were expressed relative to sh*SCN5A* cells transfected with siCTL, in the absence of any Na$_V$ inhibitor. NS stands for no statistical difference and *** denotes a statistical difference from sh*SCN5A*/siCTL at $P < 0.001$. (**d**) Cancer cell invasiveness was assessed using Matrigel-invasion chambers for MDA-MB-468 breast, H460 and A549 non-small-cell lung, and PC3 prostate cancer cells transfected with null-target siRNA (siCTL, black bar) or *SCN4B*-targeting siRNA (si*SCN4B*, red bars). Cancer cell lines known to express or not functional Na$_V$ channels are indicated as Na$_V^+$ and Na$_V^-$, respectively. The results from 3 to 12 independent experiments were presented and are expressed relative to the results obtained with the same cells transfected with siCTL. *, different from siCTL at $P < 0.05$ and *** at $P < 0.001$. Statistics presented in this figure were performed using ANOVA for multiple group comparison (**a–c**) or Student's *t*-test (**d**). All results presented in this figure are mean values ± s.e.m.

quantitative PCR (qPCR) (Supplementary Fig. 6d). In sh*SCN4B* cells, the $I_{Na}$ activation–voltage relationship was slightly depolarized as compared with shCTL cells ($V_{1/2}$–activation were $-37.2 \pm 0.9$ mV ($n = 22$) and $-40.1 \pm 1.0$ mV ($n = 18$), respectively, $P = 0.037$, Student's *t*-test). More importantly, the availability–voltage relationship was also significantly shifted to a more depolarized potential in sh*SCN4B* cells compared with shCTL cells ($V_{1/2}$–availability were $-80.6 \pm 1.6$ mV ($n = 22$) and $-86.3 \pm 1.6$ mV ($n = 18$), respectively, $P = 0.016$, Student's *t*-test; Fig. 5b), suggesting that even though there were less Na$_V$1.5 channels at the plasma membrane, they might be more active at the membrane potential of cancer cells (comprised between $-30$

and $-40$ mV) through an increased persistent window current. Indeed, while the peak $I_{Na}$ was reduced in sh*SCN4B* cells, the persistent $I_{Na}$ recorded for a membrane depolarization from $-100$ to $-30$ mV was strictly identical (Fig. 5c,d). Therefore, the ratio $I_{Na}$ persistent/$I_{Na}$ peak was significantly enhanced in sh*SCN4B* cells (Fig. 5e). The sensitivity of $I_{Na}$ to TTX was unchanged in sh*SCN4B* cells and could almost be fully inhibited by 30 μM TTX (inhibited by $85.6 \pm 2.9\%$, $n = 7$–12; Fig. 5f). Because the persistent $I_{Na}$ was similar in shCTL and sh*SCN4B* cells, there was no difference in the efflux of H$^+$ in the two cell lines, measured as previously published[25], by the addition of 130 mM NaCl in NH$_4$Cl-pulse-wash-acidified cells in a sodium-

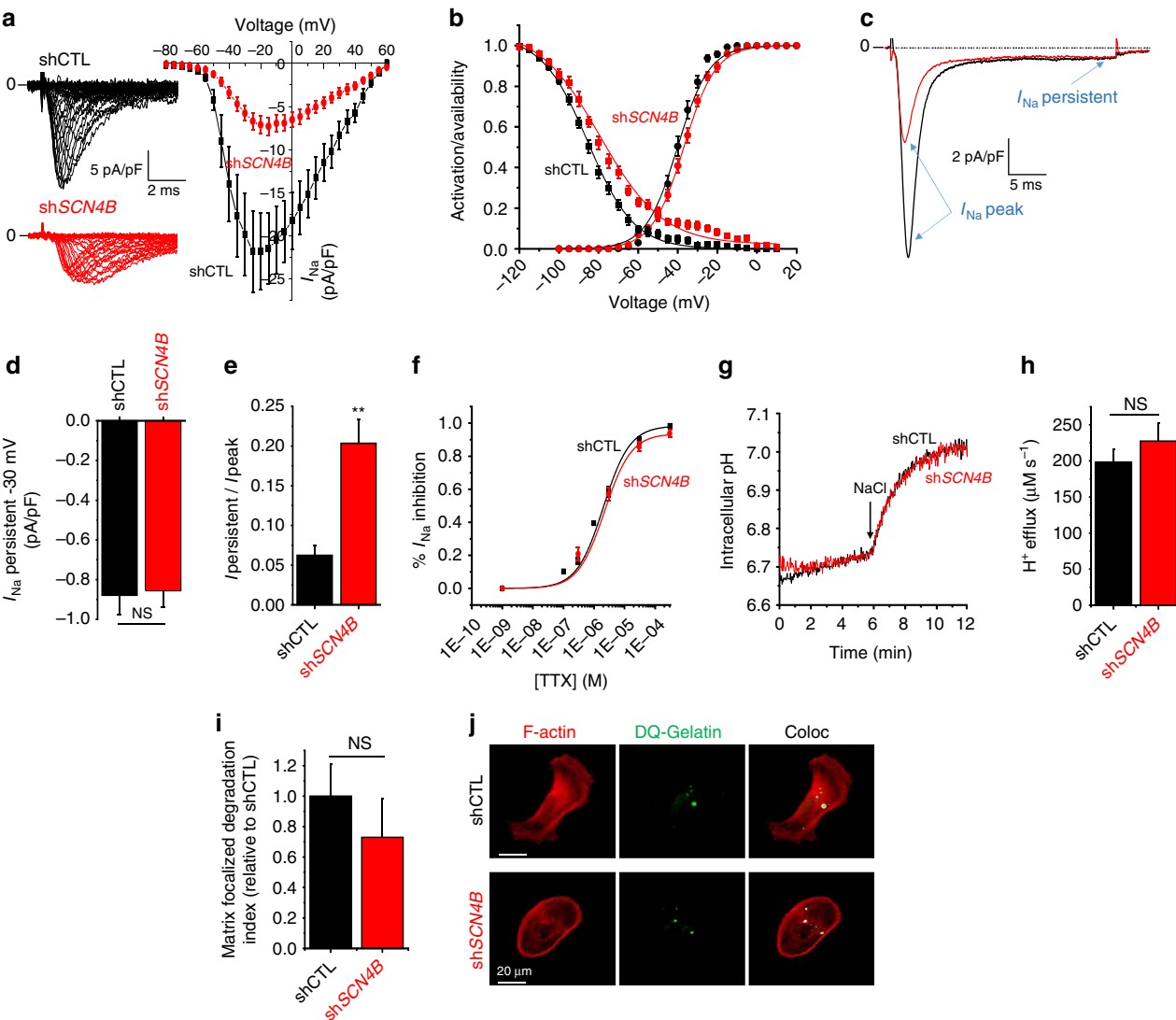

**Figure 5 | Loss of *SCN4B*/β4 expression maintains Na$_V$1.5-mediated persistent current and dependent extracellular matrix degradation.** (**a**) Sodium current ($I_{Na}$)–voltage relationships in shCTL (black squares, $n=18$) and in sh*SCN4B* (red circles, $n=22$) cells. There was a significant difference at $P<0.001$ between the two conditions in the voltage range between $-40$ and $+40$ mV. (**b**) Activation (filled circles)– and availability (filled squares )–voltage relationships in shCTL (black symbols) and sh*SCN4B* (red symbols) cells. (**c**) $I_{Na}$ peak and $I_{Na}$ persistent currents obtained from shCTL (black trace) and sh*SCN4B* (red trace) cells for a membrane depolarization from $-100$ to $-30$ mV. (**d**) Mean values ± s.e.m. of $I_{Na}$ persistent currents obtained for a membrane depolarization from $-100$ to $-30$ mV from 18 shCTL and 21 sh*SCN4B* cells. NS, not statistically different. (**e**) Mean values ± s.e.m. of $I_{Na}$ persistent/$I_{Na}$ peak currents ratios obtained from 18 shCTL and 21 sh*SCN4B* cells. **, statistically different from shCTL at $P<0.01$. (**f**) Dose–response effect of TTX on the inhibition of $I_{Na}$ peak elicited by a membrane depolarization from $-100$ to $-5$ mV in shCTL (black squares, $n=8$–12) and in sh*SCN4B* (red circles, $n=7$–12) cells. Data were fitted with the Hill equation and IC$_{50}$ values were $2.02 \pm 0.10$ and $2.24 \pm 0.11 \mu$M for shCTL and sh*SCN4B* cells, respectively. (**g**) Intracellular pH measurements using the BCECF-AM probe, in NH$_4$Cl-acidified shCTL (black trace) and sh*SCN4B* (red trace) cells in the absence of NaCl. NaCl (130 mM) was added at the time indicated (arrow). (**h**) H$^+$ efflux measurements after the addition of NaCl in conditions similar to ***g*** ($n=20$). Results are expressed as mean values ± s.e.m. NS, no statistical difference. (**i**) MDA-MB-231 shCTL or sh*SCN4B* cells were cultured on a Matrigel matrix containing DQ-Gelatin. A 'Matrix-Focalized-degradation index' was calculated as being F-actin foci (red labelling, phalloidin-Alexa594) co-localized with focused proteolytic activities (green) ($n=442$ cells for shCTL and 448 cells and sh*SCN4B*). Results are expressed as mean values ± s.e.m. NS, no statistical difference. (**j**) Representative pictures showing matrix degradation areas (green spots) and F-actin foci (red spots). Merging points (coloc), which appear as white pixels, were counted. Numbers of white pixels per cell were normalized to the mean value obtained in shCTL cells. Statistics were performed using Student's *t*-test.

free solution (Fig. 5g,h). As a consequence, sh*SCN4B* cells demonstrated identical ECM degradative activities as compared with shCTL cells (Fig. 5i,j), and similar levels of co-localization between the phosphorylated form (Y421) of the actin nucleation-promoting factor cortactin, used as a marker of invadopodial structures[41], and DQ-gelatin degradation (Supplementary Fig. 5c). These results suggested that the invadopodial activity was similar in shCTL and sh*SCN4B* cells.

**RhoA-dependent amoeboid cell migration.** While cells lacking *SCN4B* expression have the ability to degrade the ECM, the pharmacological inhibition of proteases by GM6001 (MMP inhibitor), leupeptin (cysteine, serine and threonine peptidases inhibitor) or E64 (cysteine cathepsin inhibitor) did not completely prevent the increased invasiveness observed in sh*SCN4B* as compared with shCTL cells (Fig. 6a). The most potent effect was obtained with GM6001, which reduced sh*SCN4B* cell

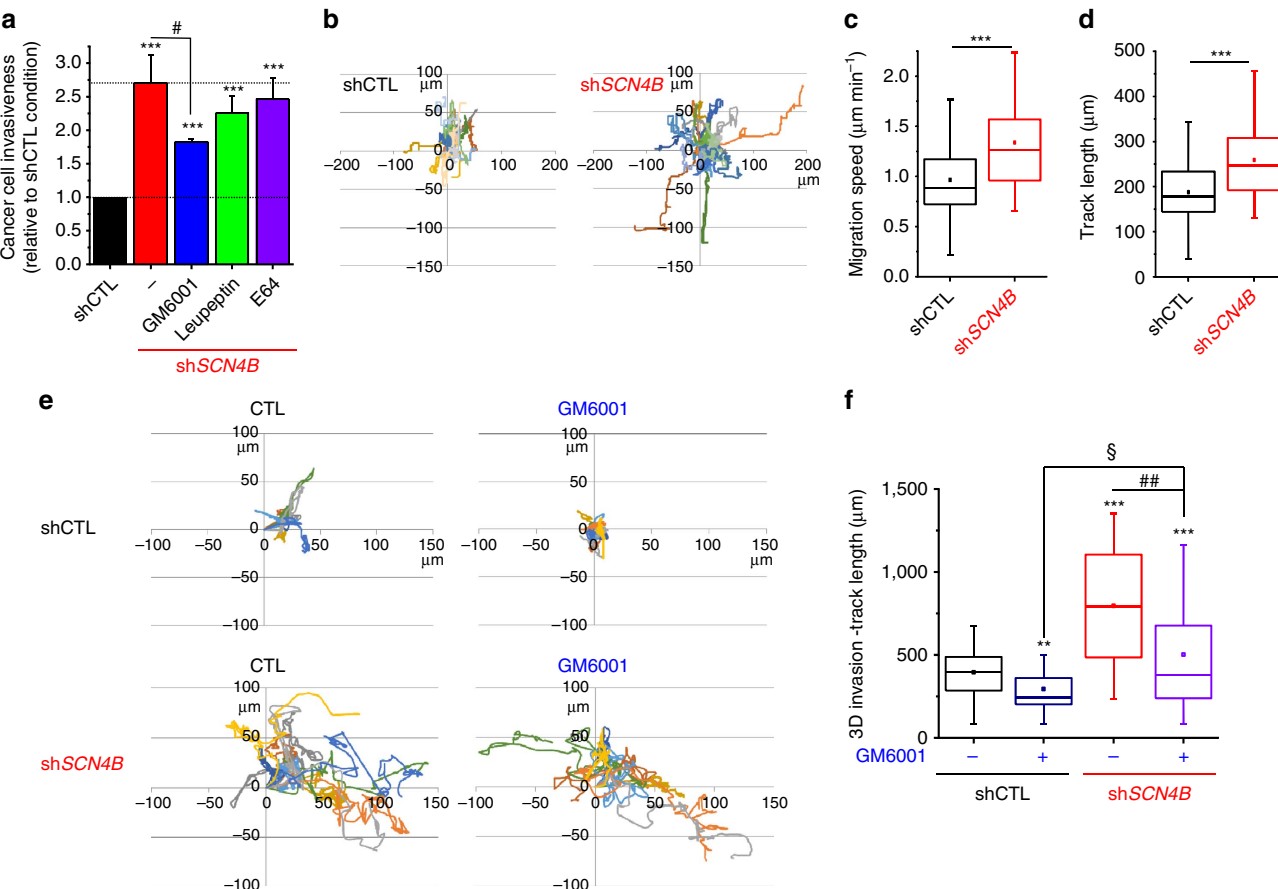

**Figure 6 | The loss of SCN4B/β4 expression promotes human cancer cell migration and invasiveness in two and three dimensions.** (a) Cancer cell invasiveness was assessed using Matrigel-invasion chambers from shCTL or shSCN4B MDA-MB-231 cells, in the absence ( − ) or presence of the protease inhibitors GM6001 (10 μM), leupeptin (200 μM) or E64 (100 μM). Results from three to seven independent experiments are presented and were expressed relative to shCTL cells in the absence of inhibitors. Results are expressed as mean values ± s.e.m. *** denotes a statistical difference from the shCTL at $P < 0.001$, and # indicates a statistical difference from shSCN4B at $P < 0.05$ (ANOVA). (b) Cancer cell migration of shCTL and shSCN4B cells measured by time-lapse microscopy to track the movement of cells over 180 min, 1 frame per min ($n = 20$ representative cells in each condition). Distances are indicated in μm. (c) The speed of migration (in μm min$^{-1}$) was analysed in shCTL and shSCN4B cells from time-lapse experiments and results shown were obtained from 106 and 96 cells, respectively. *** denotes a statistical difference from the shCTL at $P < 0.001$ (MW). (d) The track length of cell migration (in μm) was analysed over 180 min in shCTL and shSCN4B cells from time-lapse experiments and results shown were obtained from 106 and 96 cells, respectively. *** denotes a statistical difference from the shCTL at $P < 0.001$ (MW). (e) Three-dimension (3D) invasiveness of shCTL and shSCN4B cells, embedded inside Matrigel, was measured by time-lapse microscopy to track the movement of cells over 48 h (1 frame per 30 min) in the absence (CTL) or presence of the MMP inhibitor GM6001 (10 μM) ($n = 13$ representative cells in each condition). Distances are indicated in μm. (f) The track length of 3D cell invasiveness (in μm) was analysed over 48 h in shCTL and shSCN4B cells from time-lapse experiments and results shown were obtained from 30 cells in each condition. ** and *** denote statistical difference from the shCTL, CTL condition at $P < 0.01$ and $P < 0.001$, respectively. ## denotes a statistical difference from the shSCN4B, CTL condition at $P = 0.002$. § denotes a statistical difference from the shCTL, GM6001 condition at $P = 0.038$ (Dunn's test). (c,d and f) Box plots indicate the first quartile, the median and the third quartile; whiskers indicate minimum and maximum values; squares show the means. Error bars encompass 95% of data samples.

invasiveness by about 32%. This suggested that the enhancement of invasiveness was not solely due to an increase in ECM proteolysis, thus prompting us to analyse the migratory abilities of cell lines using time-lapse cell tracking experiments (Fig. 6b). As expected, the loss of SCN4B expression significantly increased the migration speed (medians were 0.889 μm min$^{-1}$ for shCTL ($n = 106$) and 1.265 μm min$^{-1}$ for shSCN4B cells ($n = 96$), $P < 0.001$, Mann–Whitney rank sum test; Fig. 6c), as well as the track length after 3 h measurements (medians were 177.76 μm for shCTL ($n = 106$) and 246.73 μm for shSCN4B cells ($n = 96$), $P < 0.001$, Mann–Whitney rank sum test; Fig. 6d), but with no change in net displacement (Supplementary Fig. 6f). This increase in migration velocity was evocative of a transition towards the amoeboid invasiveness, which is better visualized in three-dimensional (3D) matrices. To study this property more

adequately shCTL and shSCN4B cells were seeded into a 3D matrix composed of Matrigel and invasive abilities were analysed, in the absence or presence of GM6001, using time-lapse single-cell tracking over 48 h (Fig. 6e). The loss of SCN4B expression significantly increased the 3D track length travelled by cells over 48 h (medians were 399.39 μm for shCTL and 789.11 μm for shSCN4B, $P < 0.001$, Mann–Whitney rank sum test; Fig. 6f). This increase in 3D invasion could not be attributed to the increase in ECM degradative activity, since shSCN4B invasion was significantly higher than in shCTL cells in the presence of GM6001 (medians were 246.76 μm for shCTL + GM6001 ($n = 30$) and 379.87 μm for shSCN4B + GM6001 ($n = 30$), $P = 0.038$, Mann–Whitney rank sum test; Fig. 6f). The amoeboid phenotype is generally characterized by a rounded morphology, the presence of blebs at the cell surface and a relative

independence of the interaction with the substratum revealed by a decreased number of filopodial structures. Consistent with this, sh*SCN4B* cells demonstrated striking changes in morphology with a higher circularity index ($0.48 \pm 0.02$ ($n = 88$) versus $0.34 \pm 0.02$ ($n = 88$) in sh*SCN4B* and shCTL, respectively, $P < 0.001$, Students' *t*-test; Fig. 7a). Using scanning electron microscopy (SEM), we also noticed a decreased number of filopodia-like structures per cell (medians were 19.0 ($n = 60$)

versus 44.5 ($n = 66$) in sh*SCN4B* and shCTL, respectively, $P < 0.001$, Mann–Whitney rank sum test; Fig. 7b,c), and an increase in the number of blebs per cell (medians were 100.5 ($n = 82$) versus 27.5 ($n = 82$) in sh*SCN4B* and shCTL, respectively, $P < 0.001$ Mann–Whitney rank sum test; Fig. 7b–d). These blebs were relatively small with diameters between 0.5 and 0.7 μm (Fig. 7b). To confirm that the filopodia-like structures observed by SEM were real filopodia, and not retraction fibres,

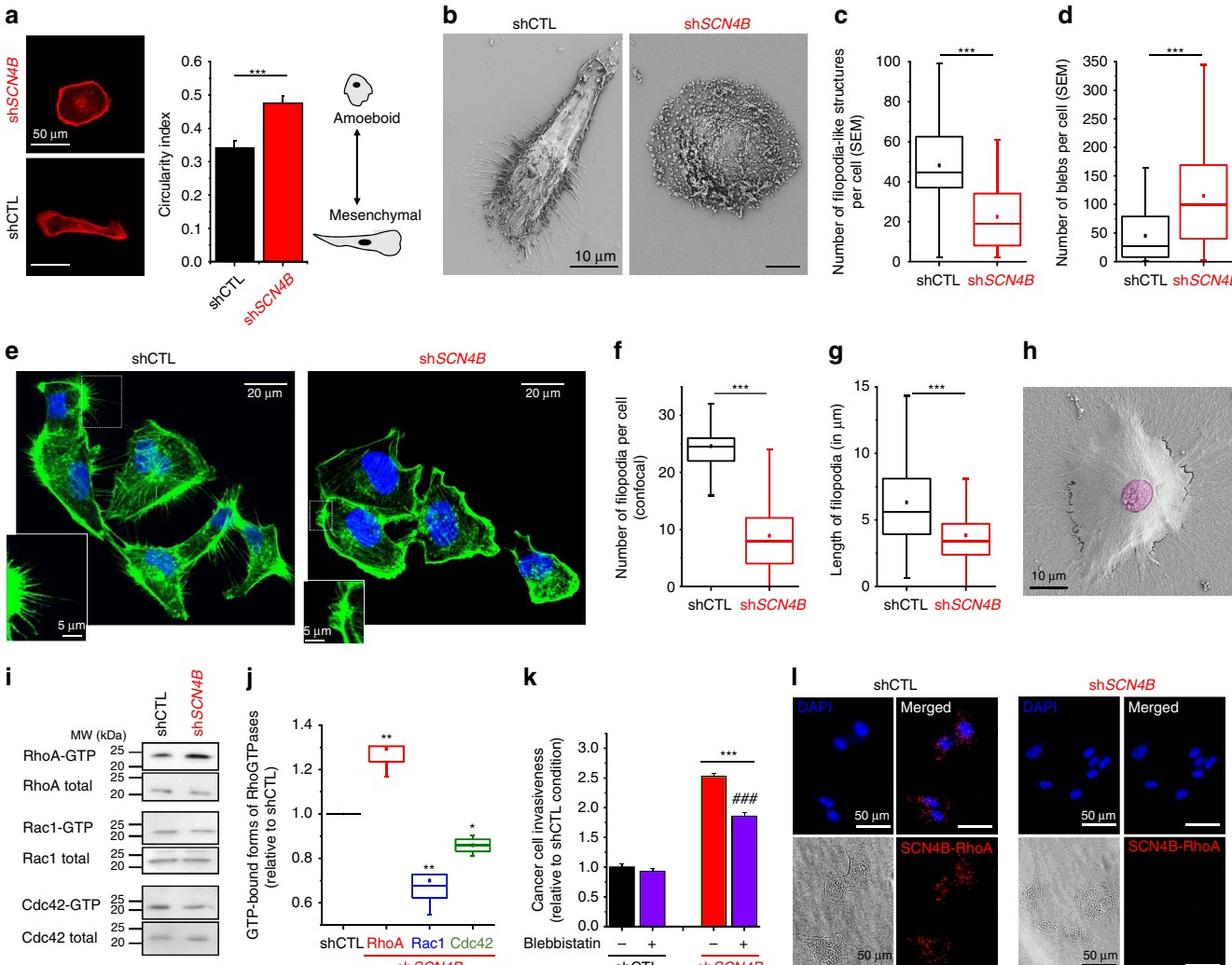

**Figure 7 | The loss of *SCN4B*/β4 expression promotes RhoA-dependent amoeboid cell transition and migration.** (**a**) F-actin was stained with phalloidin-AlexaFluor594 in shCTL and sh*SCN4B* cells and a cell circularity index was calculated ($n = 88$ cells per condition). Results are expressed as mean values ± s.e.m. ***$P < 0.001$ from shCTL (Student's *t*-test). (**b**) Representative SEM micrographs of shCTL and sh*SCN4B* cells. Scale bars, 10 μm. (**c**) Number of filopodia-like structures per cell, counted from SEM pictures in shCTL and sh*SCN4B* cells ($n = 60$ and 66 cells, respectively). ***$P < 0.001$ from shCTL (MW). (**d**) Number of blebs per cell, counted from SEM micrographs in shCTL and sh*SCN4B* cells ($n = 82$ cells per condition). ***$P < 0.001$ from shCTL (MW). (**e**) Representative confocal micrographs of shCTL and sh*SCN4B* cells for which F-actin was stained with phalloidin-AlexaFluor488 (green) and nuclei with DAPI (blue), scale bar 20 μm. For enlargements images, scale bars are 5 μm. (**f**) Number of filopodia per cell, counted from confocal micrographs in shCTL and sh*SCN4B* cells ($n = 46$ and 66 cells, respectively). ***$P < 0.001$ from shCTL (MW). (**g**) Length of filopodia, measured from confocal micrographs, in shCTL and sh*SCN4B* cells ($n = 1,070$ and 593 filopodia, respectively). ***, statistically different from shCTL at $P < 0.001$ (MW). (**h**) SEM observations of sh*SCN4B* cell invasion 24 h after cells were seeded on a layer of Matrigel (4 mg ml$^{-1}$). The coloured structure is the tip of the cell still observable above the Matrigel layer, while penetrating inside the matrix. Scale bar, 10 μm. (**i**) Western blots showing total and active GTP-bound forms of RhoA, Rac1 and Cdc42, pulled down by GST-RBD in shCTL and sh*SCN4B* cells. (**j**) Quantification of GTP-bound RhoGTPases (active), normalized to total protein level, and expressed relatively to that of shCTL ($n = 5$). **, statistically different from shCTL at $P < 0.01$ and * at $P < 0.05$ (MW). (**k**) shCTL or sh*SCN4B* cancer cell invasiveness, in the absence ($-$) or presence of blebbistatin (50 μM) ($n = 3$). Results are expressed as mean values ± s.e.m. ***$P < 0.001$ from shCTL, and ###$P < 0.001$ from sh*SCN4B* (ANOVA). (**l**) Left panel, *in situ* proximity ligation assays showing a strong proximity between *SCN4B*/β4proteins and RhoA in shCTL cells (red dots, left panel) and the absence of any proximity signal in sh*SCN4B* cells (right panel). Scale bars, 50 μm. (**c,d,f,g,j**) Box plots indicate the first quartile, the median and the third quartile; whiskers indicate minimum and maximum values; squares show the means. Error bars encompass 95% of data samples.

we also stained F-actin and performed confocal imaging (Fig. 7e). In agreement with results previously obtained in SEM, sh*SCN4B* cells displayed less filopodia than shCTL cells (median were 8 ($n = 66$ cells) versus 24.5 ($n = 46$ cells) filopodia/cells for sh*SCN4B* and shCTL, respectively, $P < 0.001$, Mann–Whitney rank sum test; Fig. 7e,f). Furthermore, filopodia in sh*SCN4B* cells were smaller than in shCTL cells (median length were 3.43 versus 5.61 µm in sh*SCN4B* and shCTL, respectively, $P < 0.001$, Mann–Whitney rank sum test; Fig. 7e–g). The morphological changes observed in sh*SCN4B* cells might result from a transition towards an amoeboid phenotype that could confer cancer cells the ability to squeeze and migrate through small gaps of the ECM. Using SEM, we confirmed that sh*SCN4B* cells demonstrated the ability to migrate through Matrigel, probably after they opened a narrow interstice by focalized degradation (Fig. 7h, Supplementary Fig. 7). The interconversions between mesenchymal and amoeboid modes of invasion are known to be orchestrated by RhoGTPases and the amoeboid movement mainly relies on the RhoA-ROCK-pMLCII signalling pathway. We therefore assessed the proportion of active (GTP-bound) RhoGTPases in shCTL and sh*SCN4B* cells by pull-down assays (Fig. 7i) and found a significant increase in RhoA concomitant with decreases in Rac1 and Cdc-42 activities (Fig. 7j). Furthermore, the inhibition of myosin II with blebbistatin[42] significantly reduced sh*SCN4B* cancer cell invasiveness by approximately 27%, while it had no effect on shCTL cell invasiveness (Fig. 7k). Interestingly, proximity ligation assays indicated a close association of the *SCN4B*/β4 protein and RhoA in shCTL cells while no signal was observed in sh*SCN4B* cells (Fig. 7l). Because the reduced expression of *SCN4B* increases cancer cell aggressiveness, we investigated whether its stable experimental overexpression (oe*SCN4B*) could have opposite effects. The overexpression of *SCN4B* gene, confirmed by qPCR and western blotting experiments (Supplementary Fig. 8a,b), significantly reduced cancer cell invasiveness by 51.6 ± 6.8% ($n = 6$) in oe*SCN4B* as compared with control cells and by about 82% as compared with sh*SCN4B* cells (Fig. 8a). To further investigate the regulation of Na$_V$ activity and Na$_V$-dependent invasiveness, we performed invasion experiments in the presence or absence of 30 µM TTX. While TTX reduced the invasion of control (oeCTL) cells to an extent similar to that found in wild-type or shCTL cells (i.e., a reduction by 30.7 ± 4.0%, $n = 6$), it had no further effect on reducing the invasive properties of oe*SCN4B* cells (Fig. 8b). Overexpressing *SCN4B* did not affect cancer cell growth and viability (Supplementary Fig. 8c). These results suggested that overexpression of *SCN4B* not only reduced the invasiveness via the *SCN4B*/β4 protein-dependent signalling pathway but also inhibited the Na$_V$-dependent mesenchymal invasion. To test this hypothesis, we analysed $I_{Na}$ in oe*SCN4B* cells. The maximal peak $I_{Na}$ was higher in oe*SCN4B* cells as compared with oeCTL or shCTL cells, while the sensitivity to TTX remained unchanged (Fig. 8c and Supplementary Fig. 8d). In oe*SCN4B* cells, the $I_{Na}$ activation–voltage relationship was not different from that of oeCTL cells ($V_{1/2}$–activation were $-42.8 ± 0.4$ mV ($n = 43$) and $-41.9 ± 0.8$ mV ($n = 15$), respectively; Fig. 8d) and the availability–voltage relationship was slightly, but not significantly, shifted towards hyperpolarized values in oe*SCN4B* as compared with those obtained with oeCTL cells ($V_{1/2}$–availability were $-88.6 ± 1.0$ and $-86.7 ± 0.5$ mV, respectively, $P = 0.459$, Mann–Whitney rank sum test; Fig. 8d). This result was not statistically significant when compared with the $V_{1/2}$–availability in shCTL cells ($-86.3 ± 1.6$ mV, $P = 0.146$, Mann–Whitney rank sum test), but was significant when compared with the $V_{1/2}$–availability in sh*SCN4B* cells ($-80.6 ± 1.6$ mV, $P < 0.001$, Mann–Whitney rank sum test). We however measured a decrease in the persistent $I_{Na}$ current

(Fig. 8e) and in $I_{Na}$ persistent/peak ratio in oe*SCN4B* cells as compared with oeCTL (Fig. 8f) and a significant reduction of the focalized Matrigel degradation (Fig. 8g), suggesting that overexpression of *SCN4B*/β4 protein slightly reduced the persistent window current and its associated ECM proteolytic (mesenchymal) activity in cancer cells. This observation was also accompanied with reductions in the cell circularity index (from 0.49 ± 0.02 ($n = 73$) to 0.38 ± 0.02 ($n = 73$) in oeCTL and oe*SCN4B*, respectively, $P < 0.001$, Student's *t*-test; Fig. 8h and Supplementary Fig. 9a), in the migration speed (medians are 0.983 µm min$^{-1}$ for oeCTL ($n = 47$) and 0.520 µm min$^{-1}$ for oe*SCN4B* cells ($n = 47$), $P < 0.001$, Mann–Whitney rank sum test; Fig. 7i) and in RhoA activity (Fig. 8j,k). Overall, these results indicate that *SCN4B*/β4 protein reduces mesenchymal invasion and prevents amoeboid migration by reducing Na$_V$ and RhoA activities, respectively.

*SCNxB*/β proteins, initially characterized as being 'auxiliary' subunits of Na$_V$, have been proposed to also act as cell adhesion molecules (CAMs) via their immunoglobulin-like extracellular domain[14]. To investigate the participation of *SCN4B*/β4 protein domains in the phenotypical changes observed, we constructed different variants of the protein intended to be overexpressed in sh*SCN4B* cells. For this purpose, the nucleotide sequence was mutated so that the transcripts would not to be targeted by the shRNA. Eight nucleotides were substituted without altering the amino acid sequence. The 'full-length' sequence allowed the expression of a wild-type β4 protein in sh*SCN4B* cells. We also constructed two truncated variants: an N-terminus-truncated protein (from residue M1 to T161), called 'ΔN-ter', containing the transmembrane and C-terminus intracellular domains of the *SCN4B*/β4 protein but completely devoid of the Ig-like extracellular domain, and a *SCN4B*/β4 protein truncated in the C-terminus region, from residue K185, and identified as being 'ΔC-ter' (Fig. 9a). The ΔC-ter construct contained the same eight substituted nucleotides with conservation of the amino acid sequence so that it could be expressed in sh*SCN4B* cells. The protein expression level of these variants at the plasma membrane was similar (Supplementary Fig. 9b). The reintroduction of the full-length *SCN4B*/β4 protein significantly reduced cancer cell invasiveness as compared with the empty vector (pSec), and as such, performed as an effective rescue. Interestingly, the ΔC-ter variant, which possessed the extracellular domain, was completely ineffective, whereas the ΔN-ter variant reduced cell invasiveness to the same extent as the full-length protein (Fig. 9b). The invasiveness of sh*SCN4B* cells overexpressing the ΔN-ter variant or the full-length protein was still superior to the one of shCTL cells by about 50%. In agreement with these results, only the full-length and the ΔN-ter proteins reduced the speed of migration and RhoA activity (Fig. 9c,d) in sh*SCN4B* cells, therefore demonstrating that the intracellular C-terminus of the *SCN4B*/β4 protein, and not the extracellular Ig-like domain, is needed to inhibit invasiveness in breast cancer cells.

**Tumour progression.** Two important steps in tumour progression are the entry of cancer cells into the vasculature and their dissemination into the general circulation before colonization of secondary organs. This progression requires cellular intravasation/extravasation abilities that can be assessed *in vitro* using transendothelial invasion (with an initial migration through the ECM, before crossing over the endothelium) and migration protocols, respectively. In both experiments, the downregulation of *SCN4B* importantly potentiated the capacity of cancer cells to migrate through a monolayer of endothelial cells, after having invaded (invasion), or not (migration), a layer of matrigel, as compared with shCTL or oe*SCN4B* cells (Fig. 10a,b).

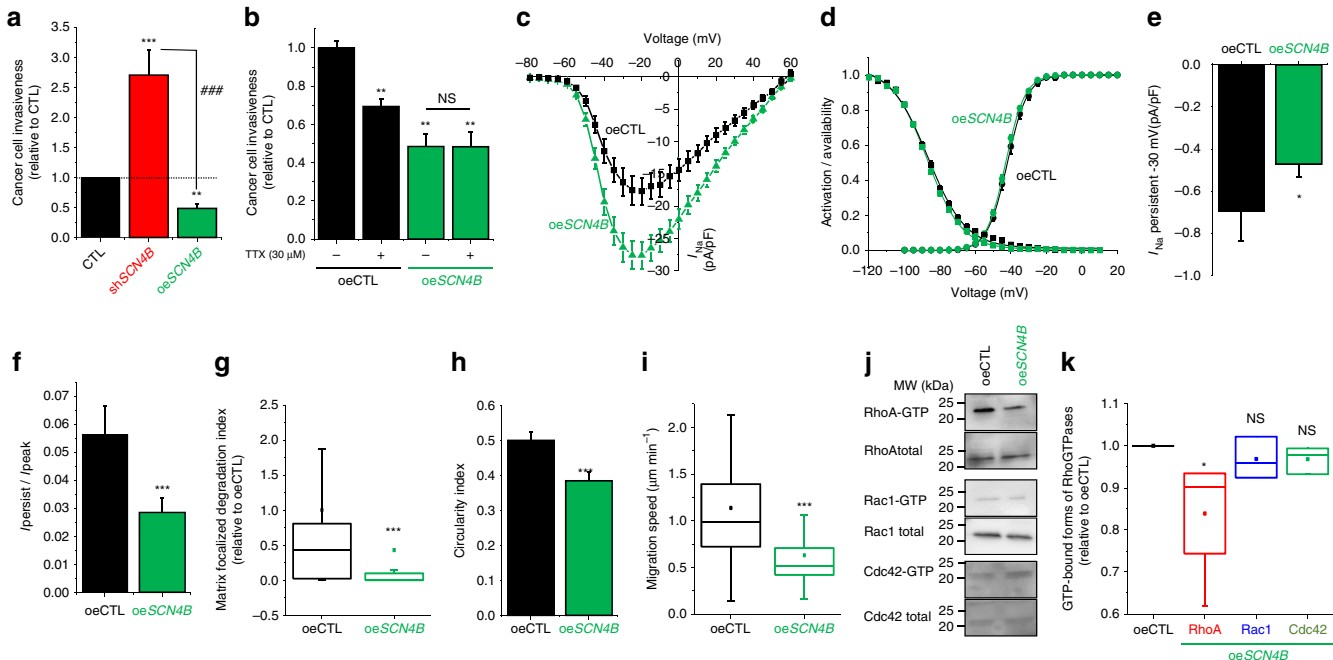

**Figure 8 | SCN4B/β4 protein overexpression inhibits cancer cell invasiveness. (a)** CTL, shSCN4B and oeSCN4B cancer cell invasiveness, in the absence ( − ) or presence of TTX (30 µM), expressed relative to oeCTL cells in the absence of TTX ($n = 6$). Results are expressed as mean values ± s.e.m. ***, different from CTL at $P < 0.001$, ** at $P < 0.01$. ###, different from shSCN4B at $P < 0.001$ (ANOVA). NS, no statistical difference. **(b)** Cancer cell invasiveness ($n = 6$) from oeCTL and oeSCN4B cells, in the absence ( − ) or presence of TTX (30 µM), expressed relative to oeCTL cells in the absence of TTX. Results are expressed as mean values ± s.e.m. **, different from oeCTL at $P < 0.01$. NS, no statistical difference (ANOVA). **(c)** $I_{Na}$–voltage relationships in oeCTL (black squares, $n = 15$) and oeSCN4B (green triangles, $n = 43$) cells. There was a significant difference at $P < 0.05$ between the two conditions in the voltage range between − 45 and + 45 mV. **(d)** Activation (filled circles)– and availability (filled squares)–voltage relationships obtained in the same oeCTL (black symbols) and oeSCN4B (green symbols) cells as in **c**. **(e)** Mean values ± s.e.m. of $I_{Na}$ persistent currents obtained for a membrane depolarization from − 100 to − 30 mV from 15 oeCTL and 43 oeSCN4B cells. *$P < 0.05$ from oeCTL (Student's t-test). **(f)** Mean values ± s.e.m. of $I_{Na}$ persistent/$I_{Na}$ peak current ratios in same conditions as in **e**. ***$P = 0.001$ (Student's t-test). **(g)** oeCTL or oeSCN4B cells were cultured on a Matrigel-composed matrix containing DQ-Gelatin, and a 'Matrix-Focalized-degradation index' was calculated ($n = 77$ and 69 cells for oeCTL and oeSCN4B, respectively). ***, statistically different from oeCTL at $P < 0.001$ (MW). **(h)** Cell circularity index was calculated from oeCTL and oeSCN4B cells ($n = 73$ cells per condition). Results are expressed as mean values ± s.e.m. ***, statistically different from oeCTL at $P < 0.001$ (Student's t-test). **(i)** Speed of migration (µm min$^{-1}$) of oeCTL and oeSCN4B cells analysed from time-lapse experiments ($n = 47$ per condition). ***, statistically different from oeCTL at $P < 0.001$ (MW). **(j)** Western blots showing total and active GTP-bound forms of RhoA, Rac1, Cdc42, pulled down by GST- in oeCTL and oeSCN4B cells. **(k)** Quantification of GTP-bound RhoGTPases in oeSCN4B cells, normalized to its total protein level, and expressed relatively to that of oeCTL cells ($n = 4$). *, statistically different from the oeCTL at $P < 0.05$. NS, no statistical difference (MW). **(g,i,k)** Box plots indicate the first quartile, the median and the third quartile; whiskers indicate minimum and maximum values; squares show the means. Error bars encompass 95% of data samples.

We then studied the involvement of the *SCN4B* gene and its expression product, the *SCN4B*/β4 protein, in tumour progression. For this we used oe*SCN4B* and sh*SCN4B* cells as models of cancer cells, coming from the same lineage and solely differing by the presence of high levels of *SCN4B*/β4 protein (similar to non-cancerous mammary cells, oe*SCN4B* cells) or the complete absence of *SCN4B*/β4 protein (similar to high-grade cancers, sh*SCN4B* cells). We developed two *in vivo* models in immuno-depressed mice. In the first model, we assessed the importance of *SCN4B* expression in human breast cancer cells for the colonization of organs. Sh*SCN4B* or oe*SCN4B* cells, both having identical *in vitro* growth and viability properties and similarly expressing the luciferase gene (Supplementary Fig. 8c–e), were injected in the tail vein of NMRI nude mice. After 9 weeks, there was no statistical difference in animal body weights between the two experimental groups (Supplementary Fig. 8f). Mice were euthanized and the isolated organs (lungs, brain, liver, bones from spine/ribs and legs) were analysed *ex vivo* for bioluminescent imaging after luciferin injection (Fig. 10c). In the sh*SCN4B* group, all mice showed lung metastases (7/7), while in the oe*SCN4B* group, only one mouse out of eight had lung colonization and there was no bioluminescent signal in other organs (Yate's $\chi^2$,

$P = 0.0041$). Altogether, there was a strong reduction in lung colonization by cancer cells overexpressing *SCN4B*/β4 compared with non-expressors (Fig. 10d). In the other model of orthotopic xenograft of mammary cancer, in which sh*SCN4B* or oe*SCN4B* cells were injected into the mammary fat pad of NOD SCID mice, the primary tumour growth was analysed as a function of time for 23 weeks. There was no statistical difference in the evolution of animal body weights between the two experimental groups (8 mice/group; Supplementary Fig. 10a). The growth of primary mammary tumours, measured with a calliper (Supplementary Fig. 10b) or by bioluminescent imaging (Fig. 10e,f), was reduced in mice implanted with oe*SCN4B* cells compared with those implanted with sh*SCN4B*. At the end of the study, primary mammary tumours from the two groups were analysed and the non-expression of *SCN4B*/β4 in the sh*SCN4B* group or expression in the oe*SCN4B* group was confirmed by immuno-histochemistry (Fig. 10g). Finally, 5/8 mice implanted with sh*SCN4B* cells had more metastatic foci (3/8 in lungs, 1 in spine and 1 in posterior legs) compared with those implanted with oe*SCN4B* cells (only 1/8 mouse showed lung metastases) (Yates $\chi^2$ $P$-value = 0.039; Supplementary Fig. 10c–e). The presence of human breast cancer cells in mouse lungs was confirmed by

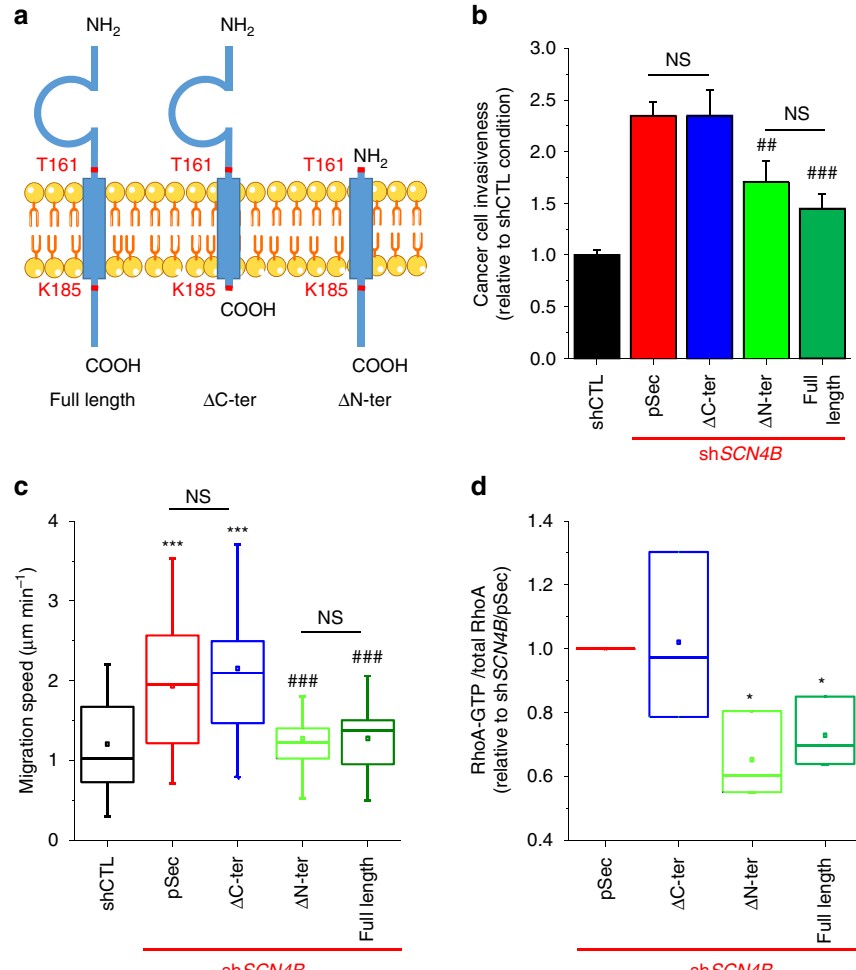

**Figure 9 | SCN4B/β4 protein inhibits cancer cell invasiveness through its intracellular C-terminus but not through its extracellular Ig-like domain.**
(**a**) Cartoon showing the transmembrane structure of the β4 protein, encoded by the *SCN4B* gene. The extracellular domain contains an Ig-like structure. After introduction of synonymous nucleotide substitutions in the *SCN4B* sequence, we have generated a sequence that is not recognized by the small hairpin RNA targeting *SCN4B* expression. This sequence has been inserted into a pSec expression vector in order to overexpress the full-length β4 protein (called 'Full-length') in sh*SCN4B* cells. Alternatively, we have also created truncated versions of the β4 protein: one containing a deletion of its intracellular C-terminus, from residue K185, and called 'ΔC-ter', and one containing a deletion of its extracellular N-terminus up to residue T161, and called 'ΔN-ter'. The nucleotide sequences were inserted into the pSec mammalian expression vector. (**b**) Cancer cell invasiveness was assessed using Matrigel-invasion chambers from shCTL and sh*SCN4B*, transfected with an empty expression vector (pSec), or transfected with 'ΔN-ter', 'ΔC-ter' or 'Full-length' encoding sequences. Results are expressed as mean values ± s.e.m. ###, statistically different from sh*SCN4B*/pSec at $P < 0.001$ and ## at $P < 0.01$. NS, not statistically different (ANOVA). (**c**) The speed of migration (in μm min$^{-1}$) was analysed from time-lapse experiments with shCTL and sh*SCN4B* cells, transfected with an empty expression vector (pSec), or transfected with 'ΔN-ter', 'ΔC-ter' or 'Full-length' encoding sequences, and results shown were obtained from 30 cells in each condition. ***, statistically different from shCTL at $P < 0.001$. ###, statistically different from sh*SCN4B*/pSec at $P < 0.001$. NS, not statistically different (Dunn's test). (**d**) Quantification of GTP-bound RhoA in sh*SCN4B* cells, transfected with empty vector (pSec), with 'ΔN-ter', 'ΔC-ter' or 'Full-length' encoding sequences. The activity of GTP-bound (active) RhoAGTPase was normalized to its total protein level, and was expressed relatively to that of sh*SCN4B*/pSec cells ($n = 3$). *, statistically different from sh*SCN4B*/pSec at $P < 0.05$ (MW). (**c**,**d**) Box plots indicate the first quartile, the median and the third quartile; whiskers indicate minimum and maximum values; squares show the means. Error bars encompass 95% of data samples.

human cytokeratin 7 immunoreactivity (Fig. 10h). All together, these results indicate that the overexpression of *SCN4B* in cancer cells reduces primary tumour growth and metastatic colonization.

## Discussion

*SCNxB*/β proteins were initially isolated from rat brain along with pore-forming Na$_V$ sodium channels[43]. They all exhibit an extracellular N-terminus containing an immunoglobulin domain. With the exception of the soluble β1B protein[44], all *SCNxB*/β subunits have a single α-helical transmembrane domain and a short intracellular domain[45]. *SCNxB*/β proteins were demonstrated to interact, through covalent or non-covalent

associations, with pore-forming Nav channels[46–49], to regulate their trafficking to the plasma membrane, as well as their biophysical[50] and pharmacological[51–53] properties. Because of these functions, *SCNxB*/β proteins were initially characterized as being auxiliary subunits of pore-forming Nav in excitable cells. Besides these canonical roles, it has later been proposed that they might also have other specific cellular functions[54]. Indeed, the presence of an Ig motif in their extracellular domain, similar to that of CAMs such as integrins, cadherins and selectins, argued for a possible function in cell adhesion properties[55,56]. *SCN1B*/β1 and *SCN2B*/β2 subunits were demonstrated to form both *trans*-homophilic and *trans*-heterophilic cell–cell and cell–matrix adhesions in cells expressing Nav, such as neurons in which

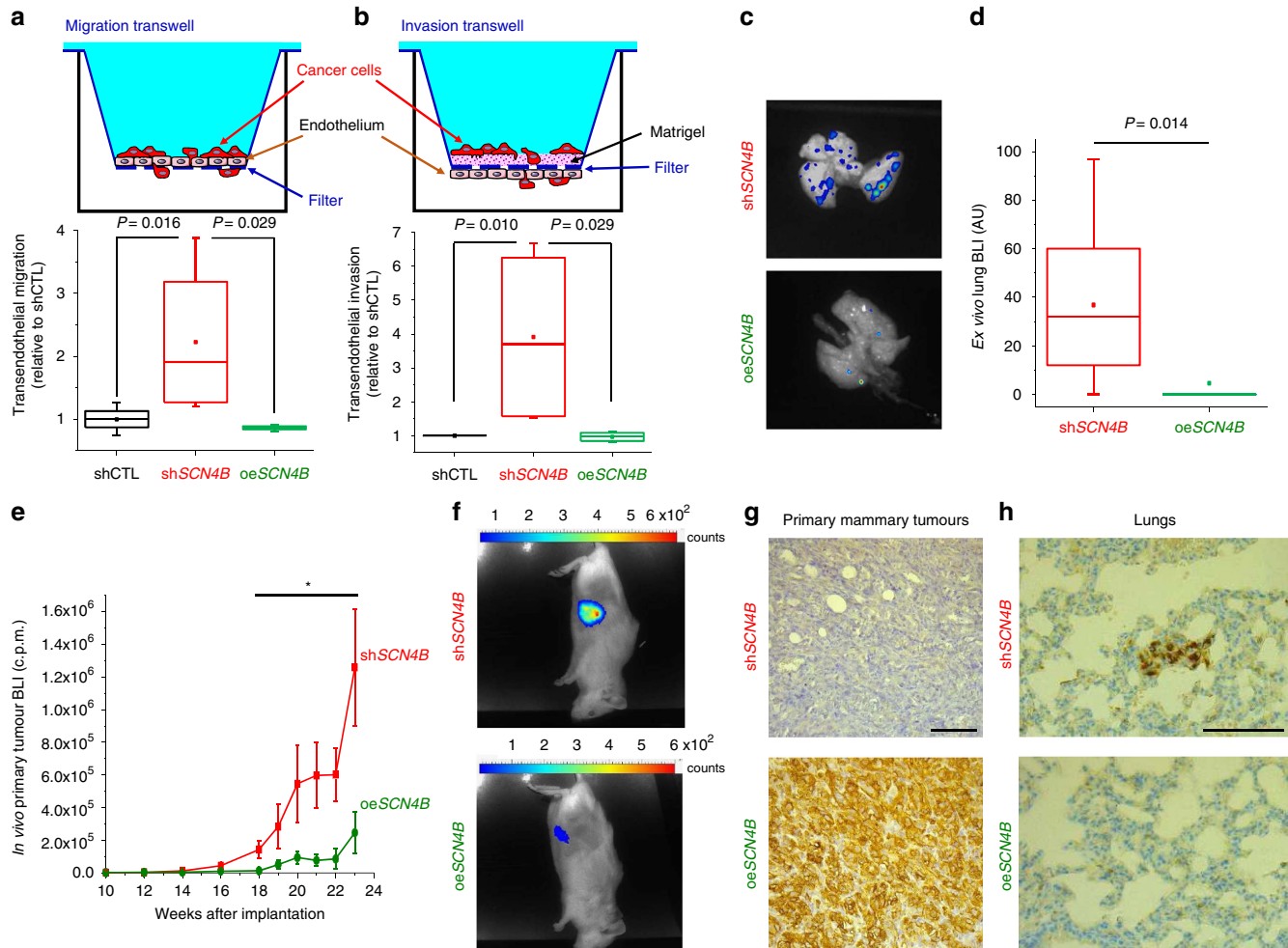

**Figure 10 | *SCN4B* expression inversely correlates with primary tumour growth and metastatic development.** (**a**) Top cartoon, transendothelial migration experiment. Bottom box plot, quantification of the number of shCTL, sh*SCN4B* or oe*SCN4B* cancer cells migrating through the endothelium (HUVEC monolayer) and the 8-µm pore-sized filter of the migration transwell, expressed as a relative number to shCTL (4 independent experiments). (**b**) Top cartoon, transendothelial invasion experiment. Bottom box plot, quantification of the number of shCTL, sh*SCN4B* or oe*SCN4B* cancer cells migrating through the extracellular matrix (matrigel) coating the 8-µm pore-sized filter of the invasion transwell, then endothelium (HUVEC monolayer), expressed as a relative number to shCTL (three independent experiments). (**c**) Bioluminescent imaging (BLI) performed in NMRI nude mice tail vein-injected with MDA-MB-231-Luc cells that do not express (sh*SCN4B*), or which overexpress, the *SCN4B* protein (oe*SCN4B*). Representative *ex vivo* lung BLI, after organ isolation, at completion of the study (ninth week after cell injection). (**d**) BLI quantification of excised lungs from mice injected with sh*SCN4B* cells ($n = 7$) and mice injected with oe*SCN4B* cells ($n = 8$) (MW). (**e**) Mean ± s.e.m. *in vivo* BLI value of tumours (expressed in c.p.m.) as a function of time recorded in the whole body of mice. * denotes a statistical difference from the sh*SCN4B* group at $P < 0.05$ (Student's *t*-test). (**f**) Representative bioluminescent images of mammary tumours in sh*SCN4B* and oe*SCN4B* experimental groups (23rd week after cell implantation). (**g**) Immunohistochemical analyses of primary mammary tumours obtained from same mice as in Fig. 8e,f implanted with sh*SCN4B* cells (top image) or with oe*SCN4B* cells (bottom image). Slides were counterstained with haematoxylin (blue labelling), and incubated with anti-mouse *SCN4B*/β4 antibodies and immunohistochemistry was performed using the streptavidin-biotin-peroxidase method with diaminobenzidine as the chromogen (brown labelling). Scale bars, 100 µm. (**h**) Immunohistochemical analyses of lungs obtained from the same mice as in Fig. 6e,f, implanted with human sh*SCN4B* cells (top image) and oe*SCN4B* cancer cells (bottom image). Slides were counterstained with haematoxylin (blue labelling), and human breast cancer cells were identified using anti-human cytokeratin 7 immunohistochemical (brown) labelling. Scale bars, 100 µm. (**a,b,d**) Box plots indicate the first quartile, the median and the third quartile; whiskers indicate minimum and maximum values; squares show the means. Error bars encompass 95% of data samples.

they were proven to be critical for neurites outgrowth, axonal fasciculation and interactions with glial cells[14].

The expression of *SCNxB*/β proteins and their physiological role in non-excitable cells have not been characterized. Their participation in oncogenic processes was not studied as much as that of pore-forming Nav proteins[17]. The most studied isoform in cancer is *SCN1B*/β1, and results published so far could appear contradictory. An initial *in vitro* study indicated that the expression of *SCN1B*/β1 was higher in poorly invasive than in highly invasive breast cancer cells. In weakly invasive MCF-7

cells, the downregulation of *SCN1B*/β1 reduced cell adhesion, and increased cell migration. Correlatively, its overexpression in MDA-MB-231 cells, increased cell process length and adhesion, reduced lateral motility and cell proliferation[33]. These results suggested that *SCN1B*/β1 expression could prevent cancer cell invasion. In contrast, a recent study demonstrated that *SCN1B*/β1 was overexpressed in breast cancer biopsies, compared with non-cancer breast samples. Furthermore, the overexpression of *SCN1B*/β1 in breast cancer cells potentiated their invasiveness *in vitro*, and increased primary tumour growth and metastatic

development *in vivo*[27]. The proinvasive role of *SCN1B*/β1 was proposed to depend on both Nav-dependent and Nav-independent mechanisms, relying on the extracellular Ig domain of the protein[27]. In this study, we found that *SCN1B* gene is downregulated in breast cancer compared with normal breast tissues. Nevertheless, we identified *SCN1B* as being the *SCNxB* gene the most highly expressed in cancer cells. Interestingly, the downregulation of *SCN1B*/β1 reduces *in vitro* invasiveness of MDA-MB-231, which is in line with results obtained by Nelson *et al.*[27] showing this isoform displays pro-invasive roles *in vitro*. Concerning the *SCN4B*/β4 protein, there was no information regarding its potential involvement in the oncogenic process. Only one study identified that *SCN4B* expression was decreased in cervical cancer biopsies compared with non-cancer samples[57].

Aggressive cancer cells present an important plasticity enabling them to use different invasion modes, conferring an ability to adapt to their microenvironment, matrix composition, meshwork, eventually promoting tumour growth and metastases development. While RhoGTPases are involved in invasive phenotypes, the upstream mechanism regulating their respective activation is not known. Here, we identified the *SCN4B* gene as an important modulator of RhoA activity and invasiveness in cancer cells. The *SCN4B*/β4 protein was known to be an auxiliary subunit of Na$_V$ in excitable cells. The present study shows for the first time that this protein is expressed in non-cancer epithelial cells, which do not express Na$_V$, suggesting a physiological non-auxiliary function. Furthermore, the expression of *SCN4B*/β4 is reduced in cancer tissues, and more specifically when tumours gain invasive properties (transition from grade I to grade II). *SCN4B*/β4 is almost absent in high-grade tumours and metastases. At the cellular level, the loss of *SCN4B*/β4 in cancer cells further stimulates their invasiveness by favouring amoeboid migration, supported by the overactivation of the RhoA pathway, yet keeping the ECM-degradative activity intact. Conversely, the overexpression of *SCN4B*/β4 reduces cancer cell invasiveness, tumour growth and metastatic progression, supporting our proposal that the *SCN4B* gene is a metastasis-suppressor gene.

This study also demonstrates that the *SCN4B*/β4 protein acts both as Na$_V$ pore-subunit regulator and as RhoGTPases regulator in cancer cells. Indeed, the suppression of *SCN4B* gene expression in cancer cells reduced the peak sodium current due to Na$_V$1.5 activity. This could be attributed to a reduction of *SCN5A* expression, and to a consequent reduction of channel density at the plasma membrane. However, we found that channels expressed at the plasma membrane of cancer cells show a higher activity through an increased persistent window current. The loss of interaction between Na$_V$1.5 and *SCN4B*/β4 protein might therefore be responsible for a gain-of-function of Na$_V$1.5. This is similar to what was observed in the case of the cardiac long QT syndrome LQT10, in which a missense mutation in the *SCN4B* gene, resulting in the L1759F amino acid substitution in the β4 protein, induces a gain-of-function of Na$_V$1.5 with an increased persistent sodium current[29]. Because of this regulation of Na$_V$1.5, the loss of *SCN4B* expression in cancer cells left Na$_V$-dependent mesenchymal invasiveness unaffected, through the maintenance of a persistent sodium current that regulates ECM proteolysis, but also increased Na$_V$-independent amoeboid-related cell migration, through the dynamic regulation of RhoGTPases. This latter property is under the control of the intracellular C-terminus and is independent of the extracellular CAM domain. These results contrast with those obtained with the pro-invasive *SCN1B*/β1 for which the extracellular Ig domain is crucial for its CAM function in breast cancer cells[27]. Our results further indicate that the intracellular C-terminus of *SCN4B*/β4 might play a critical role in maintaining epithelial function and integrity.

In conclusion, this study shows that the loss of *SCN4B* gene expression in cancer cells promotes the acquisition of an amoeboid–mesenchymal hybrid phenotype, while its overexpression reduces both mesenchymal and amoeboid invasive capacities. The expression level of *SCN4B* could therefore represent an important prognosis marker in cancers from epithelial origin.

## Methods

**Prognostic analyses of gene expression in breast cancers.** Analyses were performed using the software Breast Cancer Gene-Expression Miner v3.0 (bc-GenExMiner v3.0; http://bcgenex.centregauducheau.fr) developed by the Integrated Center of Oncology René Gauducheau (Nantes-Saint Herblain, France), based on DNA microarrays results collected from published cohorts[58]. Briefly, several statistical tests were conducted on each cohort and on all cohorts pooled together with data from all studies previously converted to a common scale with a suitable normalization. The prognostic impact of each gene was evaluated by means of the univariate Cox proportional hazards model. Results were displayed by cohorts and pools and are illustrated in a forest plot. Kaplan–Meier curves were then obtained on the pool with the gene values dichotomized according to gene expression median (calculated from the pool). Cox results corresponding to dichotomized values were displayed on the graph.

***In silico* RNA expression analyses.** Expression of *SCNxB* genes in cancer tissues was studied using the web-based 'The Cancer Genome Atlas' (http://cancergenome.nih.gov) from the US National Cancer Institute and the National Human Genome Research Institute that integrates RNE sequences databases. Data are expressed as RPKM (Reads Per Kilobase per Million).

**Immunohistochemistry.** The degree of *SCN4B*/β4 protein expression in normal and dysplastic mammary tissues, as well as in mammary ductal and lobular carcinomas, was analysed by standard immunohistochemistry procedures. Tissue microarrays (TMA) from formalin-fixed, paraffin-embedded mammary tissues were purchased from US Biomax Inc. (ref. BR1003, BC081120, BR10010a; Rockville, USA), comprising normal, hyperplastic and dysplastic mammary samples, lobular and ductal mammary carcinomas (separated in well (grade I), moderately (grade II) and poorly (grade III) differentiated carcinomas) and LNM samples. Similar analyses were also performed in a lung TMA (LC951; US Biomax Inc) containing normal lung, cancer lung (grades IA to IIIB) tissues and metastases (in lymph nodes, bones and intestine). TMA were analysed in blind conditions by C.M.-C. (anatomopathologist) and P.P. (immunologist, Clinical University Hospital Virgen de la Arrixaca, Murcia, Spain). Normal breast and cancer biopsies from the University-Hospital of Tours were coming from the tumour collection declared to the French Ministry of Research (No. DC2008-308) and were prepared by R.G. and analysed by G.F. (clinical anatomothologist at the University-Hospital of Tours, France).

Briefly, after deparaffinization and rehydration, sections were treated with a high-pH (Tris buffer/EDTA, pH 9.0) target retrieval procedure (Dako PT-link; Dako, USA). Endogenous peroxidase was then blocked by a commercial solution (Dako REAL), and incubated overnight with a 1/100 dilution of the primary polyclonal rabbit antibody anti-*SCN4B*/β4 protein (HPA017293; Sigma-Aldrich, France) at 4 °C. The antibody used recognizes the following extracellular peptide sequence of the β4 protein: LLPCTFSSCFGFEDLHFRWTYNSSDAFKILIEGTVK NEKSDPKVTLKDDDRITLVGSTKEKMNNISIVLRDLEFSDTGKYTCHVKNPK ENNLQHHATIFLQVV. Sections were then incubated with a commercial anti-rabbit-labelled polymer (Dako EnVision FLEX; Dako) for 30 min at RT. Immunoreaction was finally revealed with 3-3′ diaminobenzidine solution (Dako) for 5 min. Positive reaction was identified by a cytoplasmic dark-brown precipitate. To determine the degree of protein expression in tissues, a qualitative scale was used, for negative ( − ), weak ( + ) and strong ( + + ) cytoplasmic expression. A unique score was given per core. The number of samples is representative of the number of cores, which is equivalent to the number of patients.

Lung and primary tumours from *in vivo* mouse experiments were fixed in formalin, included in paraffin, and cut in 5 µm tissue sections. Slides were deparaffinized, rehydrated and heated in citrate buffer pH 6 for antigenic retrieval. The primary antibodies included monoclonal anti-human cytokeratin 7 (clone OV-TL 12/30; Dakocytomation, Glostrup, Denmark, dilution 1/100, 1 h), used on lung sections, and polyclonal anti-mouse *SCN4B*/β4 (dilution 1/100, 1 h), used on primary tumour sections. Immunohistochemistry was performed using the streptavidin-biotin-peroxidase method with diaminobenzidine as the chromogen (Kit LSAB, Dakocytomation). Slides were finally counterstained with haematoxylin. Negative controls were obtained after omission of the primary antibody or incubation with an irrelevant antibody.

**Reagents and antibodies.** TTX was purchased from Latoxan (France), and A803467 from R&D systems (France). Fluorescent probes and conjugated

antibodies were purchased from Fisher Scientific (France). Drugs and chemicals were purchased from Sigma-Aldrich.

**Cell culture and cell lines.** All cell lines were from the American Type Culture Collection (LGC Promochem, France) and were grown at 37 °C in a humidified 5% $CO_2$ incubator. The immortalized normal mammary epithelial cells MCF-10A were cultured in Dulbecco's modified Eagle's medium (DMEM)/Ham's F-12, 1:1 mix containing 5% horse serum (Invitrogen, France), 10 µg ml$^{-1}$ insulin, 20 ng ml$^{-1}$ epidermal growth factor, 0.5 µg ml$^{-1}$ hydrocortisone and 100 ng ml$^{-1}$ cholera toxin. MCF-7, MDA-MB-468 and MDA-MB-435s breast cancer cells were cultured in DMEM supplemented with 5% fetal calf serum (FCS). PC3 prostate, H460 and A549 non-small-cell lung cancer cells were cultured in DMEM supplemented with 10% FCS. MDA-MB-231-Luc human breast cancer cells, stably expressing the luciferase gene[22], were cultured in DMEM supplemented with 5% FCS. We constructed a lentiviral vector encoding a short hairpin RNA (shRNA) specifically targeting human SCN5A transcripts[35]. The sequence encoding shSCN5A, inhibiting the expression of Na$_V$1.5 protein, was obtained by DNA polymerase fill-in of two partially complementary primers: 5′-GGATCCCCAAGGCACAAGTGCGTGCG CAATTCAAGAGA-3′ and 5′-AAGCTTAAAAAAAGGCACAAGTGCGTGCG CAATCTCTTGAA-3′.

Similarly, we constructed a lentiviral vector encoding a short hairpin RNA (shRNA) specifically targeting human SCN4B transcripts, inhibiting the expression of β4 protein. The sequence encoding the shSCN4B was obtained by DNA polymerase fill-in of two partially complementary primers; this method also allowed the introduction of two restriction enzyme sites to facilitate manipulations. Forward primer: shβ4-BamHI 5′-GGATCCCCCAGCAGTGACGCATTCAAGA TTCTTCAAGAGA-3′and reverse primer: shβ4-HindIII 5′-AAGCTTAAAAACA GCAGTGACGCATTCAAGATTCTCTCTTGAA-3′. We also constructed a lentiviral vector expressing a null-target shRNA (pLenti-shCTL), using the following primers: 5′-GGATCCCCGCCGACCAATTCACGGCCGTTCAAGAG ACG-3′ and 5′-AAGCTTAAAAAGCCGACCAATTCACGGCCGTCTCTTGA ACG-3′. We constructed an expression plasmid containing the human SCN4B coding sequence to overexpress β4. This sequence was synthetized by Proteogenix (France) and inserted in pcDNA3.1, using the following sequences: 5′-GGATCC GCCGCCACC-3′ and 5′-GCGGCCGCCTCGAG-3′. We designed the mutated sequences coding for the 'Full-length SCN4B/β4' and truncated proteins, which were then synthetized by Proteogenix (France) and all sequences obtained were inserted into pSecTag2 hygro B vector (V910-20; Invitrogen) with the In-fusion HD cloning Plus kit (Clontech). We constructed a plasmid containing the sequence coding for the N-terminally truncated (from residue 1 to residue T161, 'ΔN-ter') protein containing the transmembrane and C-terminal intracellular domain of the SCN4B/β4 protein. The sequence was obtained by PCR elongation using two specific primers: forward primer 5′-GCGCCGTACGAAGCTGACCT GGAGTTCAGCGAC-3′ and reverse primer 5′-ACACTGGAGTGGATCTCAC ACTTTTGAAGGTGGTT-3′. Similarly, an SCN4B/β4 protein truncated in the C-terminus, starting with residue K185 and identified as being 'ΔC-ter' was designed. Importantly, for 'ΔC-ter' and 'full-length' the nucleotide sequence was mutated by substitution to avoid targeting by the shRNA targeting native SCN4B gene. The protein sequence remained unaffected. The sequence targeted by the shRNA was 5′-CAGCAGTGACGCATTCAAATTC-3′, while the substituted untargeted sequence was 5′-TAGTAGCGATGCCTTTAAATAC-3′. All variants exhibited an His-Tag and their expression and subcellular localization, after transfection of shSCN4B cells, were assessed by epifluorescence imaging using a primary rabbit anti-HisTag antibody (T2767; Invitrogen), and a secondary goat anti-rabbit Texas Red antibody (Thermofisher, France).

Mycoplasma contamination tests were performed every week (Lonza, MycoAlert Mycoplasma Detection Kit).

**Reverse transcription of RNA and real-time PCR.** Total RNA extraction was performed using the RNAgents Total RNA Isolation System (Promega, France). RNA yield and purity were determined by spectrophotometry and only samples with an A260/A280 ratio above 1.6 were kept for further experiments. Total RNA were reverse-transcribed with the RT kits Ready-to-go You-prime First-Strand Beads (Amersham Biosciences, UK). Random hexamers pd(N)$_6$ 5′-phosphate (0.2 µg; Amersham Biosciences) were added and the reaction mixture was incubated at 37 °C for 60 min. Real-time PCR experiments were performed as previously described[24]. Primers sequences are given in Supplementary Table 1.

**siRNA transfection and efficacy assessment.** MDA-MB-231-Luc human breast cancer cells were transfected with 20 nM siRNA targeting the expression of SCN1B (siSCN1B, sc-97849), SCN2B (siSCN2B, sc-96252), SCN4B (siSCN4B, sc-62982) or scrambled siRNA (siCTL, siRNA-A sc-37007), which were produced by Santa Cruz Biotechnology and were purchased from Tebu-Bio (France). Transfection was performed with Lipofectamine RNAi max (Invitrogen) according to the manufacturer's instructions, and used 24 h after transfection. The efficiency of siRNA transfection was verified by qPCR using an iCycler system (BioRad, USA) and by western blotting.

**Western blotting experiments.** Cells were washed with phosphate-buffered saline (PBS) and lysed in the presence of a lysis buffer (50 mM Tris, pH 7, 100 mM NaCl,

5 mM $MgCl_2$, 10% glycerol, 1 mM EDTA), containing 1% Triton-X-100 and protease inhibitors (Sigma-Aldrich). Cell lysates were cleared by centrifugation at 16,000 g for 10 min. Western blotting experiments were performed according to standard protocols. Total protein concentrations were determined using the Pierce BCA Protein Assay Kit Thermoscientific (Fisher Scientific, France). Protein sample buffer was added and the samples were boiled at 100 °C for 3 min. Total protein samples were electrophoretically separated by sodium dodecyl sulfate-poly-acrylamide gel electrophoresis in 10% gels, and then transferred to polyvinylidene fluoride membranes (Millipore, USA). The SCNxB/β proteins were detected using anti-SCN1B/β1 (1/1,000, AV35028; Sigma-Aldrich), anti-SCN2B/β2 (1/200, HPA012585; Sigma-Aldrich) and anti-SCN4B/β4 (1/1,000, HPA01293, Sigma-Aldrich) rabbit polyclonal primary antibodies and horseradish peroxidase-conjugated goat anti-rabbit IgG secondary antibody at 1:2,000 (TebuBio, France). HSC70 protein was detected as a sample loading control using anti-HSC70 mouse primary antibody at 1:30,000 (TebuBio) and horseradish peroxidase -conjugated anti-mouse-IgG secondary antibodies at 1:2,000 (TebuBio). Proteins were revealed using electrochemiluminescence-plus kit (Pierce ECL Western Blotting Substrate; Fisher Scientific) and captured on Kodak Bio-Mark MS films (Sigma-Aldrich). Full scans of western blots are shown in Supplementary Fig. 11.

**RhoGTPases pull-down assays.** Pull-down assays were performed according to the manufacturer's protocol (Cat#BK030, RhoA/Rac1/Cdc42 Activation Assay Combo Biochem Kit; Cytoskeleton, Inc.). Briefly, cells were washed on ice with ice-cold PBS, then lysed and scraped with ice-cold cell lysis buffer containing protease inhibitors (Sigma-Aldrich). Cell lysates were clarified by centrifugation at 10,000 g, +4 °C, 1 min, and the supernatant was snap-frozen in liquid nitrogen. Ten microlitres of clarified cell lysate were used to perform the protein assay (Cat#23225, BCA protein assay; Thermofisher). Total proteins (300 µg) were incubated with 10 µg of PAK-PDB beads or 30 µg Rhotekin-RBD beads for 1 h on a rotator at +4 °C. Samples and beads were washed with 500 µl washing buffer, resuspended in Laemmli buffer and boiled for 2 min prior to performing western blotting experiments. Antibodies for RhoA, Rac1 and Cdc42 were provided with the kit and used according to the manufacturer's protocol.

**Cellular electrophysiology.** Patch pipettes were pulled from borosilicate glass to a resistance of 3–5 MΩ. Currents were recorded, in whole-cell configuration, under voltage-clamp mode of the patch-clamp technique, at room temperature, using an Axopatch 200B patch clamp amplifier (Axon Instrument, USA). Analogue signals were filtered at 5 kHz, and sampled at 10 kHz using a 1,440A Digidata converter. Cell capacitance and series resistance were electronically compensated by about 60%. The P/2 sub-pulse correction of cell leakage and capacitance was used to study Na$^+$ current ($I_{Na}$). Sodium currents were recorded by depolarizing the cells from a holding potential of −100 mV to a maximal test pulse of −30 mV for 30 ms every 500 ms. The protocol used to build sodium current–voltage ($I_{Na}$–$V$) relationships was as follows: from a holding potential of −100 mV, the membrane was stepped to potentials from −80 to +60 mV, with 5-mV increments, for 50 ms at a frequency of 2 Hz. Availability–voltage relationships were obtained by applying 50 ms prepulses using the $I_{Na}$–$V$ curve procedure followed by a depolarizing pulse to −5 mV for 50 ms. In this case, currents were normalized to the amplitude of the test current without a prepulse. Conductance through Na$^+$ channels ($g_{Na}$) was calculated as already described[21]. Current amplitudes were normalized to cell capacitance and expressed as current density (pA/pF). The Physiological Saline Solution had the following composition (in mM): NaCl 140, KCl 4, $MgCl_2$ 1, $CaCl_2$ 2, D-glucose 11.1 and HEPES 10, adjusted to pH 7.4 with NaOH (1 M). The intrapipette solution had the following composition (in mM): KCl 130, NaCl 15, $CaCl_2$ 0.37, $MgCl_2$ 1, Mg-ATP 1, EGTA 1, HEPES 10, adjusted to pH 7.2 with KOH (1 M).

**Measurement of intracellular pH.** Cells were incubated for 30 min at 37 °C in Hank's medium containing 2 µM BCECF-AM (2′,7′-bis-(2-carboxyethyl)-5-(and-6)-carboxyfluorescein; excitation 503/440 nm; emission 530 nm). Excess dye was removed by rinsing the cells twice with Physiological Saline Solution. H$^+$ efflux was measured as previously described[25,59].

**Cell viability.** Cells were seeded at $4 \times 10^4$ cells per well in a 24-well plate and were grown for a total of 5 days. Culture media were changed every day. Viable cell numbers were assessed by the tetrazolium salt assay as previously described[24] and normalized to the appropriate control condition (MDA-MB-231-Luc or shCTL).

**In vitro invasion assays.** Cell invasiveness was analysed as previously described[25] using culture inserts with 8-µm pore size filters covered with Matrigel (Becton Dickinson, France). Briefly, the upper compartment was seeded with $6 \times 10^4$ cells in DMEM supplemented with 5% FCS. The lower compartment was filled with DMEM supplemented with 10% FCS (or 15% FCS for non-small-cell lung and prostate cancer cells), as a chemoattractant. After 24 h at 37 °C, remaining cells were removed from the upper side of the membrane. Cells that had invaded and migrated through the insert and were attached to the lower side were stained with

DAPI and counted on the whole area of the insert membrane. *In vitro* invasion assays were performed in triplicate in each separate experiment.

**3D invasion assays.** Five thousand shCTL or shSCN4B MDA-MB-231-luc cells were suspended in DMEM containing Matrigel (Corning; ref 356230, 2.7 mg ml$^{-1}$ final concentration), seeded in 96-well microplates and placed at 37 °C, 5% $CO_2$. One hundred microlitres of DMEM + 5% FBS were added in each well 30 min after seeding. Time-lapse acquisitions (1 image per 30 min for a total duration of 48 h) were performed in the presence or absence of the MMP inhibitor GM6001 (10 μM final concentration) to monitor 3D cancer cell invasiveness and started 4 h after cell seeding.

**Epifluorescence imaging.** For the assessment of ECM degradation, cells were cultured for 24 h on glass coverslips coated with a matrix composed of Matrigel (4 mg ml$^{-1}$, final concentration) and containing or not DQ-gelatin (25 μg ml$^{-1}$, Thermofisher) as a fluorogenic substrate of gelatinases. They were then washed in PBS, fixed with 3.7% ice-cold paraformaldehyde in PBS. Cells were permeabilized with a solution containing 50 mM NH$_4$Cl, 1% BSA and 0.02% saponin, then saturated in a solution containing 3% BSA and 3% normal goat serum . F-actin was stained with phalloidin-AlexaFluor594. Epifluorescence microscopy was performed with a Nikon TI-S microscope and analysed using both NIS-BR software (Nikon, France) and ImageJ software 1.38I (http://rsbweb.nih.gov/ij). Pixels corresponding to the co-localization of F-actin condensation areas and focal spots of DQ-gelatin proteolysis (excitation wavelength: 495 nm, emission wavelength: 515 nm) were quantified per cell, giving a Matrix-focalized-degradation index.

Phospho (Y421)-cortactin was detected using the primary rabbit anti-pY421 cortactin antibody (Millipore) and the fluorescent-conjugated secondary anti-rabbit TexasRed antibody (Thermofisher). Pixels corresponding to the co-localization of phospho-cortactin and focal spots of DQ-gelatin proteolysis were quantified per cell using ImageJ software 1.38I.

For the assessment of cell morphology, cells were cultured for 24 h on glass coverslips and F-actin was stained with phalloidin-AlexaFluor594. A circularity index was calculated from pictures as being $4\pi \cdot area/perimeter^2$. A value approaching 0 indicates an increasingly elongated shape while a value of 1.0 indicates a perfect circle.

Proximity ligation assays were performed according to the standard protocols[26] using the Duolink-'In-cell Co-IP' kit (OLink Biosciences)[60] using anti-SCN4B and Anti-RhoA primary antibodies.

**Confocal imaging.** 5,000 cells were seeded on Labtek coverslips (Thermofisher) and placed in the incubator at 37 °C, 5% $CO_2$ for 24 h. They were then washed twice in PBS, fixed with 3.7% ice-cold paraformaldehyde in PBS. Cells were permeabilized with a solution containing 50 mM NH$_4$Cl, 1% BSA and 0.02% saponin, then saturated in a solution containing 3% BSA and 3% normal goat serum. F-actin was stained with phalloidin-AlexaFluor488 (Thermofisher). Cells were observed on a confocal microscope, at ×600 magnification (Olympus Fluoview FV500 Laser Scanning Confocal Biological Microscope) and image acquisition was performed using Fluoview 500 v.5 software (Olympus, Tokyo, Japan).

**Scanning electron microscopy.** Cells were fixed by incubation for 24 h in 4% paraformaldehyde, 1% glutaraldehyde in 0.1 M phosphate buffer (pH 7.2). They were then washed in PBS and post-fixed by incubation with 2% osmium tetroxide for 1 h. Samples were then fully dehydrated in a graded series of ethanol solutions and dried in hexamethyldisilazane (HMDS; Sigma, USA). Finally, cells were coated with 40 Å platinum, using a GATAN PECS 682 apparatus (Pleasanton, USA), before observation under a Zeiss Ultra plus FEG-SEM scanning electron microscope (Oberkochen, Germany). Cells were pseudocolored using PowerPoint software (Microsoft, USA).

**Zebrafish invasion assays.** Zebrafish (*Danio rerio*), from the Zebrafish International Resource Centre (ZIRC), were maintained in re-circulating tanks according to the standard procedures[61]. Adult fishes were maintained at 26 °C, with a light/dark cycle of 14/10 h, and were fed twice daily, once with dry flake food (PRODAC) and once with live *Artemia salina* (MC 450, IVE AQUACULTURE). Zebrafish embryos were maintained in egg water at 28.5 °C, fed for 5 days with NOVO TOM and with live *A. salina* at 11 days of life. All experiments were performed in compliance with the Guidelines of the European Union Council for animal experimentation (86/609/EU) and were approved by the Bioethical Committee of the University Hospital Virgen de la Arrixaca (Spain). The colonization of zebrafish embryos was previously described[35]. Briefly, MDA-MB-231 breast cancer cells transfected with siRNA targeting the expression of SCN4B gene (SiSCN4B) or null-target siRNA (siCTL) were trypsinized 24 h after transfection, washed and stained with the vital cell tracker red fluorescent CM-Dil (Vibrant; Invitrogen). Fifty labelled cells were injected into the yolk sac of dechorionated zebrafish embryos using a manual injector (Narishige). Fish with fluorescently labelled cells appearing outside of the implantation area at 2 h post-injection were excluded from further analysis. All other fishes were incubated at 35 °C for 48 h and analysed with a SteReo Lumar V12 stereomicroscope equipped

with an AxioCam MR5 camera (Carl Zeiss). The evaluation criteria for embryos being colonized by human cancer cells was the presence of more than five cells outside of the yolk sac. A zebrafish (ZF) colonization index was calculated as being the proportion of embryos being colonized (by at least five human cancer cells) in the siSCN4B condition divided by the proportion of invaded embryos in the siCTL condition.

**Transendothelial migration/invasion.** Human umbilical vein endothelial cell (HUVEC; Promocell, Germany) were cultured in medium 199 (1X; Gibco) containing 20% FCS, 100 μg ml$^{-1}$ heparin and 50 μg ml$^{-1}$ of endothelial cell growth supplement (Sigma-Aldrich) and were used up to the fifth passage. For transendothelial migration, 100,000 HUVEC were seeded in the upper side of 8-μm-pore sized transwell migration inserts (Corning, France) and grown for 6 days. Endothelial culture medium was carefully removed and replaced by DMEM 10% FCS in the lower chamber and DMEM 5% FCS containing 60,000 MDA-MB-231 in the upper chamber. MDA-MB-231 cells were stained with 4 ng μl$^{-1}$ CM-Dil (Thermo Fisher Scientific) prior seeding.

For transendothelial invasion, matrigel-coated transwell invasion inserts were inverted in order to plate the bottom surface with 100,000 HUVEC for 4 h at 37 °C, 5% $CO_2$. Invasion chambers were then inverted again, inserted into a 24-well plates and HUVEC were cultured for 6 days in endothelial culture medium before plating MDA-MB-231 cells in the upper chamber. MDA-MB-231 cells were CM-Dil stained as described above. After 18 h of culture, transwells were washed in PBS and fixed in formaldehyde 4% for 20 min.

Transendothelial electrical resistance measurements were recorded every day for 6 days until the transendothelial resistance was stable (>10 kΩ) on the same transwells using an EVOM2 epithelial Voltohmmeter (World Precision Instruments, France).

**In vivo tumour models.** All animals were bred and housed at the *In Vivo* platform of the Cancéropôle Grand Ouest at Inserm U892 (Nantes, France) under the animal care license no. 44278. The project was approved by the French national ethical committee (APAFIS 00085.01).

Unanaesthetized six-week-old female NMRI Nude Mice (Charles River Laboratories) were placed into a plastic restraining device, and $2 \times 10^6$ MDA-MB-231-Luc cells (shSCN4B, 7 mice/oeSCN4B, 8 mice) suspended in 100 μl PBS were injected into the lateral tail vein through a 25-gauge needle as previously described[22]. At necropsy, *ex vivo* BLI measurement for each collected organ was performed within 15 min of D-luciferin intraperitoneal injection (150 mg kg$^{-1}$). Photons emitted by cancer cells were counted by bioluminescent imaging (ΦimagerTM; BIOSPACE Lab) and expressed in counts per minute (c.p.m.).

Six-week-old female Rag2 −/− Il2rg −/− mice (NOD SCID; Charles River Laboratories) were injected into the sixth right inguinal mammary fat pad with $2 \times 10^6$ MDA-MB-231-Luc cells (shSCN4B or oeSCN4B, 8 mice per group) suspended in 100 μl PBS while under isoflurane anaesthesia. Tumour growth was monitored by bioluminescence imaging[22]. Animal weight was measured every week or every other week, and the primary tumour volume (mm$^3$) was measured with a calliper and calculated as length × height × width (in mm). Mice were euthanized 24 weeks following implantation of tumour cells and metastatic bioluminescence was measured[22].

**Statistical analyses.** Statistical analyses on immunohistochemistry staining were performed using the Yate's $\chi^2$ test using the online interactive Chi-square test software Quantpsy (http://www.quantpsy.org/chisq/chisq.htm). Other data were displayed as mean ± s.e.m. and $n$ = sample size, and were analysed using parametric statistical tests (Student's *t*-test or ANOVA) when they followed a normal distribution and equal variances. Alternatively, when samples did not follow a normal distribution, or when variances failed to be comparable, data were displayed as box plots indicating the first quartile, the median and the third quartile, and squares for comparison of means. In these cases, adequate non-parametric statistical tests were used (Mann–Whitney rank sum tests, Dunn's tests, ANOVA on ranks). Statistical analyses were performed using SigmaStat 3.0 software (Systat software Inc.) and statistical significance is indicated as *$P < 0.05$; **$P < 0.01$ and ***$P < 0.001$. NS stands for not statistically different.

**Data availability.** Expression of SCNxB genes in cancer tissues was studied using the web-based 'The Cancer Genome Atlas' (http://cancergenome.nih.gov). The authors declare that all other data supporting the findings of this study are available within the paper and its Supplementary Information files or available from the authors upon request.

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

## Acknowledgements

This work was supported by the 'Ministère de la Recherche et des Technologies', the Inserm, the 'Ligue Nationale Contre le Cancer–Interrégion Grand-Ouest', the Région

Centre (grant 'Na$_V$Metarget', project 'ARD2020 Biomédicaments') and the 'Association CANCEN'. J.L.G.-P. was supported by 'Le Studium'. We thank Mrs Catherine Le Roy for secretary and administrative assistance, and Dr Armelle Vincemeux for her help with IHC staining of β4 in human breast normal tissues. We thank Prof. Stephan J. Reshkin (University of Bari, Italy), Dr Lin-Hua Jiang (University of Leeds, UK) and Dr Philippe Chavrier (Institut Curie, France) for their critical reading of the manuscript.

## Author contributions

All authors contributed extensively to the work presented in this study. E.B. performed and analysed immunofluorescence imaging, time-lapse microscopy experiments, assessed cell adhesion and transendothelial migration. E.B., V.D. and F.G. performed cell culture, molecular and cellular biology experiments, assessed cell viability, migration and invasion. E.B., V.D., M.A. and M.-L.C. performed zebrafish experiments. I.D. participated to cell culture. C.M.-C. and P.P. performed and analysed immunohistochemical experiments on tissue microarrays, and R.G. and G.F. on mice tissues. S.M.-L. and T.O. performed mice experiments, E.B., S.R., S.C. and P.B. analysed *in vivo* data. E.B., V.D., E.P. and A.M. participated to lentiviral particles production and the generation of small hairpin RNA or overexpressing cancer cell lines. S.R performed electrophysiology experiments. E.B., F.G. and J.B.-G. performed scanning electron microscopy experiments. J.B.-G. and S.R. performed *in silico* expression analyses. P.G.F. contributed in discussion and correction of the manuscript. S.R., S.C. and P.B. obtained research grants, directed the research, designed the study, analysed the data and wrote the manuscript.

## Additional information

**Competing financial interests:** The authors declare no competing financial interests.

