## [Peer Review File · Nature Communications]

Reviewer #1 (Remarks to the Author)

In this manuscript, Bon and colleagues show for the first time that the sodium channel beta-4 subunit is downregulated in metastatic breast cancer cells. Using the MDA-MB-231 metastatic breast cancer cell line model, they show that suppressing beta-4 expression with RNAi increases cell migration, invasion, tumour growth and metastasis, independent of channel activity. The authors provide further evidence to suggest that beta-4 elicits its effect on migration via inhibition of RhoA activation. This is an important area of investigation and it is an interesting and technically sound study. I have a few comments and suggestions for consideration regarding additional controls and further interpretations of the data that I think would improve the manuscript.

1. In the last sentence of the Introduction, the authors state that the participation of non-pore-forming beta subunits has not been studied as extensively as for alpha, and their roles during metastatic progression remain largely unknown. I agree with this comment in principle. However, I think that it would be worth mentioning and discussing in this manuscript the work of Nelson et al (Int J Cancer 135, 2338-2351, 2014), who have previously shown that the closely-related beta-1 subunit promotes tumour growth and metastasis in the MDA-MB-231 model. This prior work is particularly interesting in the context of the present study, because Bon and colleagues show in supplementary Figure 1A that SCN1B is downregulated in breast cancer vs. normal breast tissue, which contrasts with the observations of Nelson et al. Even more interesting is that both Bon and colleagues (Figure 2F) and Nelson et al (2014) independently show that SCN1B/beta-1 promotes in vitro invasion in MDA-MB-231 cells. Thus, the two studies agree in terms of in vitro invasion, but disagree in terms of in silico analysis of expression array data. These agreements/contrasts between the studies should be mentioned and possible explanations discussed. An important difference between beta-4 and beta-1 seems to be that the Ig domain of beta-1 is critical for its function in MDA-MB-231 cells (Nelson et al 2014), whereas beta-4 function is dependent only on its intracellular domain. This difference is suggestive and interesting and could also be discussed!

2. Please could the authors indicate what peptide sequence was used to raise the beta-4 antibody used in Figure 1. Was it from the extracellular terminus or the intracellular domain? Would it recognise full-length and cleaved forms? Given the authors' very interesting data regarding sufficiency of the intracellular domain for beta-4 function in MDA-MB-231 cells, and that beta subunits are proteolytically processed (Wong et al (2005) J Biol Chem 280, 23009-23017), the part(s) of the beta-4 subunit that are detected by this antibody in tissues could have important implications for interpretation of the staining. On the other hand, if information on the epitope/peptide used to make this antibody is not available (e.g. for commercial reasons), the authors should state this in the methods.

3. One possible concern with interpretation of the zebrafish colonisation is the time taken for the in vivo experiment and the possibility that the siRNA (which is a transient manipulation) is no longer functional by the time the cells invade in vivo. Is it possible to confirm functionality of the siRNA at the end of the colonisation assay, e.g. by extraction of cells and QPCR? Such evidence would clearly strengthen conclusions of this experiment.

4. With regard to the electrophysiology and invasion data in Figure 3, a challenging observation to interpret is that when SCN4B is knocked down, there is no effect on persistent current, but the effect of TTX on invasion seems to be lost. Therefore, how/why does TTX have no effect in shSCN4B cells given that the channel is still functional and carrying a persistent current? The situation gets even more complex/interesting when considering the overexpression data in Figure 4, where beta-4 up-regulation seems to have the same effect as the RNAi, decoupling the effect of TTX on invasion, but at the same time, altering the current. I think these observations are important and they, together with possible explanations, should be discussed in more detail. Finally, what is the proposed mechanism for loss of transient current in the absence of beta-4 but no change in persistent current?

5. In Figure 2C, a loading control needs to be included (e.g. HSC70, as in Figure 2D). It would be helpful to also show the western indicating absence of beta-3 in MDA-MB-231 cells, together with positive and loading controls, since this result is mentioned in the text. The western in Supplementary Figure 7B also needs loading control, e.g. HSC70.

6. In Figure 3N what is the beta-4 expression status across these different cell lines? It would be helpful to include a quick QPCR experiment here, or in the supplementary data, to show whether or not beta-4 expression is similar across this cell line panel.

Minor comments:

1. The expression data in Supplementary Fig 2 appear to be presented differently (log₂ median-centred intensity) than elsewhere in the manuscript (RKPD). Why is this? The different analytical approach for Supplementary Fig 2 needs to be added to the methods.

2. The steady-state inactivation curve for shSCN4B cells in Figure 3E looks to me to be biphasic. Have the authors tried fitting to a double-Boltzmann equation, (e.g. see Lopez-Santiago et al (2006) J Neurosci 26, 7984-7994)? A two-component fit might suggest the presence of two channel types, and/or incomplete inactivation.

3. For the immunohistochemical quantification in Fig 1: was the analysis performed blinded to tumour type/outcome data? The method of blinding (or lack of it) should be mentioned in the methods.

4. In supplementary Figure 4D some other alpha subunits are downregulated at the mRNA level in shSCN4B cells. Why is this? This should be discussed.

Reviewer #2 (Remarks to the Author)

The authors investigate the role of SCN4B, which encodes for beta4 protein originally described as an auxiliary subunit of NaV channels. They show that beta2 is expressed in epithelial tissues, but that expression disappears as tumors become more aggressive. Using in vitro assays, they show that knockdown beta4 leads altered cell morphology, and increases in 2D migration speed, invasion through matrigel, and RhoA activity; overexpression of the protein reduces invasion, with all effects being independent of NaV. The authors used xenograft models to examine these phenotypes in vivo, looking at primary tumor growth and metastasis. This is an interesting paper and, in principle, could be a nice contribution to the field. The authors should perform some additional cell biological experiments to support their hypothesis that SCN4B depletion causes a shift from mesenchymal to amoeboid motility/invasion, and to characterize the cell biological effects of SCN4B a bit further using standard assays.

Major comments:

-The 2D migration and matrigel transwell invasion assays are supportive of potential amoeboid like transitions in the SCN4B depleted cells, as are the data on increased RhoA activity and decreased matrix degradation/invasion, but these are not entirely satisfying given that true amoeboid migration cannot be assessed in 2D (or in transwell matrigel assays). More compelling data would come from performing 3D migration/invasion into collagen gels +/- GM6001 to determine whether SCN4B knockdown allows for protease independent migration.

-The changes in protease activity assessed by the "focalized degradation index" are suggestive of changes in invadopodia formation/function. The authors should explore this using standard invadopodia assays and marker analysis.

- SEM cannot be used to distinguish between filopodia and retraction fibers (4F), and the F-actin

labeling in 4E does not show filopodia, which should be evident (perhaps there was insufficient exposure time or the focal plane with filopodia was missed). Filopodia should be quantified using actin labeling, preferably with filopodial markers (e.g. fascin, VASP), or by time-lapse imaging.

- IHC quantification, in Fig 1, the authors quantify levels of beta4. Does the number of samples shown in 1a represent the number of fields? Or the number of patients? Was there one score per core, or multiple per core according to field of view? Please clarify in methods

- Expression of SCN4B is restricted to epithelial tissues:

The authors write 'This suggests that SCN4B/ β 4 is normally expressed in normal epithelial cells (as confirmed by immunohistochemical analyses of normal oesophagus, (Suppl. Fig. 3)' The relevance of the staining of oesophagus tissue is not clear and not related to the rest of the study. It would be more interesting to perform co-stains for epithelial cell markers, or show staining of normal lung to show epithelial localization, or perhaps normal prostate, colon or prostate, as these data are referred to in Fig S2.

- Expression of SCN4B in cell lines (Figure 2): In Fig 2a-c, authors show mRNA levels in multiple breast cell lines. How does the protein expression compare? Are the differences seen at the mRNA level reflected in the protein? In fig 2D, does knockdown of one of the SCN genes affect expression (protein/mRNA) of the others? Please show these controls. Similarly, in Fig3, does knockdown of SCN5A affect SCN 4B levels?

- Increased association of RhoA with SCN4B (Fig 4m). How do these data fit in to the hypothesis proposed by the authors? Does SCN4B bind directly to Rho to keep it inactive or prevent its activation?

-overexpression experiments- generation of overexpressing cell lines: FigS7b- what are all the bands in the overexpression? Does overexpression lead to different isoforms or PTMs that affect MW? Why do the authors chose to to overexpress SCN4B by 40,000K fold overexpression at the mRNA level? This is likely not physiologically relevant.

Fig5H- authors should show representative image of oe cells

Fig5L-Orescue experiments: what is the level of expression for these constructs? 40,000 X? or back to normal levels. Show WB.

Please adjust language for Figure 5M in the results: the full length only rescues 50% phenotype, but not all of it.

- in vivo experiments: It is unclear why the authors chose to not show data from a control cell line in any of these experiments. Does overexpression or knockdown of SCN4B induces changes in either direction that are significantly different from control cells? It is hard to evaluate the significance of these findings without the right controls.

Fig6e- how do the authors explain the differences in primary tumor size? Given the differences in primary tumor size, the authors should quantify metastatic index (numbers of mets relative to tumor size) to measure metastatic potential.

- Scale bars: Many of the images are missing scale bars, please add these.

Please add scale bar to images in 1b and d., also S2D, 2E

Minor comments:

p4: 'The levels of β 4 expression were similar in mammary hyperplasia and dysplasia. 'Clarify- similar normal breast vs hyperplasia or dysplasia? Or similar between hyperplasia and dysplasia?

p.5 'Importantly, SCN1B and SCN4B genes appeared to be the two most highly expressed genes in non-cancer tissues (Suppl. Fig. 1g), and SCN4B the most significantly down-regulated gene in breast cancer tissues (Suppl. Fig. 1h).'

This is confusing, sounds like Scn4B is the two most highly expressed gene out of all the genes. Is this what the authors meant? Or is it within this family of genes?

Figure 4m- shctrl and shSCN4B representative images and quantification should all be next to each other in the same panel for better comparison

Figure 3D- Does overexpression of SCN5a in shSCN4B rescue the Na phenotype?

Style-

Some figures have the panel letter in capitals and some not. Please be consistence.

Reviewer 1

In this manuscript, Bon and colleagues show for the first time that the sodium channel beta-4 subunit is downregulated in metastatic breast cancer cells. Using the MDA-MB-231 metastatic breast cancer cell line model, they show that suppressing beta-4 expression with RNAi increases cell migration, invasion, tumour growth and metastasis, independent of channel activity. The authors provide further evidence to suggest that beta-4 elicits its effect on migration via inhibition of RhoA activation. This is an important area of investigation and it is an interesting and technically sound study. I have a few comments and suggestions for consideration regarding additional controls and further interpretations of the data that I think would improve the manuscript.

We sincerely thank this reviewer for all the constructive comments made on our initial manuscript which were definitely pertinent and helped us to improve the quality and relevance of our study.

1. In the last sentence of the Introduction, the authors state that the participation of non-pore-forming beta subunits has not been studied as extensively as for alpha, and their roles during metastatic progression remain largely unknown. I agree with this comment in principle. However, I think that it would be worth mentioning and discussing in this manuscript the work of Nelson et al (Int J Cancer 135, 2338-2351, 2014), who have previously shown that the closely-related beta-1 subunit promotes tumour growth and metastasis in the MDA-MB-231 model. This prior work is particularly interesting in the context of the present study, because Bon and colleagues show in supplementary Figure 1A that SCN1B is downregulated in breast cancer vs. normal breast tissue, which contrasts with the observations of Nelson et al. Even more interesting is that both Bon and colleagues (Figure 2F) and Nelson et al (2014) independently show that SCN1B/beta-1 promotes in vitro invasion in MDA-MB-231 cells. Thus, the two studies agree in terms of in vitro invasion, but disagree in terms of in silico analysis of expression array data. These agreements/contrasts between the studies should be mentioned and possible explanations discussed. An important difference between beta-4 and beta-1 seems to be that the Ig domain of beta-1 is critical for its function in MDA-MB-231 cells (Nelson et al 2014), whereas beta-4 function is dependent only on its intracellular domain. This difference is suggestive and interesting and could also be discussed!

The referee is perfectly right, and we apologize for such a rough presentation of the involvement of beta subunits in the oncogenic properties. Initially, this manuscript was sent to Nature Medicine and we were limited by the size of the manuscript and the total number of words. This was frustrating to us ! After the manuscript was rejected by Nat Med (with no external review), it was transferred to Nature Communications.

We agree with the referee that, in our initial manuscript, the study was not properly discussed and that the comparison with important results such as those brought by many teams, such as the one of Nelson and collaborators, were missing.

Our mistake is now repaired, and these important studies are now mentioned in the discussion part.

2. Please could the authors indicate what peptide sequence was used to raise the beta-4 antibody used in Figure 1. Was it from the extracellular terminus or the intracellular domain? Would it recognise full-length and cleaved forms? Given the authors' very interesting data regarding sufficiency of the intracellular domain for beta-4 function in MDA-MB-231 cells, and that beta subunits are proteolytically processed (Wong et al (2005) J Biol Chem 280, 23009-23017), the part(s) of the beta-4 subunit that are detected by this antibody in tissues could have important implications for interpretation of the staining. On the other hand, if information on the epitope/peptide used to make this antibody is not available (e.g. for commercial reasons), the authors should state this in the methods.

The peptide sequence used to raise the beta4 antibody is the shown in red in the entire beta4 sequence (entry Q8IWT1). This sequence has been added to the Methods section.

NH₂-

MPGAGDGGKAPARWLGTGLLGLFLLPVTLSEVSVGKATDIYAVNGTEILLPCTFSSCFGFEDLH
 FRWTYNSSDAFKILIEGTVKNEKSDPKVTLKDDDRITLVGSTKEKMNNISIVLRDLEFSDTGKYTC
 HVKNPKENLQHHATIFLQVVDRLLEEVDNTVTLLILAVVGGVIGLLILLIKKLIIFILKKTREKKK
 ECLVSSSGNDNTENGLPGSKAEKPPSKV
 -COOH

Signal peptide
 Transmembrane domain

The epitope recognized by the antibody is located in the extracellular domain. In the study from Wong and collaborators, it is indicated two possible cleavage sites by secretases (red arrows) in the extracellular domain of the mouse beta4 subunit ¹, which displays the following sequence.

NH₂-
 MSRAGNRGNTQARWLGTGLLGLFLLPMYLSLEVSVGKATTIYAINGSSILLPCTFSSCYGFENLYF
 KWSYNNSETSRILIDGIVKNDKSDPKVRVKDDDRITLEGSTKEKTNNISILLSDFSDTGRTYTCFV
 RNPKEKDLNNSATIFL↓QVVDKLEKVDNTVTLLILAVVGGVIGLLVCILLKLIITFILKKTREKK
 KECLVSSSGNDNTENGLPGSKAEKPPTKV
 -COOH

Signal peptide
 Transmembrane domain

While the mouse sequence is slightly different from the human one, the homology between the two sequences is high enough (90%) to consider that similar cleavage could occur in the human protein (see alignment of sequences below). Therefore, we cannot exclude the fact that the antibody used in immunohistochemistry on patient's tissues does not recognize cleaved fragment of the beta4 subunit in the extracellular domain, and solely composed of the transmembrane and intracellular domains.

Score	Expect	Method	Identities	Positives
347 bits(889)	2e- 127()	Compositional matrix adjust.	181/228(79%)	206/228(90%)
Features:				
Query	1	MPGAGDGGKAPARWLGTGLLGLFLLPVTLSEVSVGKATDIYAVNGTEILLPCTFSSCFG	60	
Sbjct	1	MSRAGNRGNTQARWLGTGLLGLFLLPMYLSLEVSVGKATTIYAINGSSILLPCTFSSCYG	60	
Query	61	FEDLHFRWTYNSSDAFKILIEGTVKNEKSDPKVTLKDDDRITLVGSTKEKMNNISIVLRD	120	
Sbjct	61	FENLYFKWSYNNSETSRILIDGIVKNDKSDPKVRVKDDDRITLEGSTKEKTNNISILLS	120	
Query	121	LEFSDTGKYTCHVKNPKENLQHHATIFLQVVDRLLEEVDNTVTLLILAVVGGVIGLLILI	180	
Sbjct	121	LEFSDTGRTYTCFVRNPKEKDLNNSATIFLQVVDKLEKVDNTVTLLILAVVGGVIGLLVCI	180	
Query	181	LLIKKLIIFILKKTREKKKECLVSSSGNDNTENGLPGSKAEKPPSKV	228	
Sbjct	181	LLLKLIITFILKKTREKKKECLVSSSGNDNTENGLPGSKAEKPPTKV	228	

However, when looking at the different results obtained at the gene expression level (DNA microarrays and *In Silico* RNA expression analyses) we observe a similar reduction of *SCN4B* expression in cancer tissues. Furthermore, our results show that the extracellular N-terminus of the beta4 protein is not involved in the regulation of cancer cell invasiveness. Therefore, we believe that the reduction of Beta4 IHC signal in cancer tissues is true and representative of the loss of *SCN4B* gene expression.

3. One possible concern with interpretation of the zebrafish colonisation is the time taken for the in vivo experiment and the possibility that the siRNA (which is a transient manipulation) is no longer functional

by the time the cells invade in vivo. Is it possible to confirm functionality of the siRNA at the end of the colonisation assay, e.g. by extraction of cells and QPCR? Such evidence would clearly strengthen conclusions of this experiment.

The referee is right, and even though we recorded a statistical difference in zebrafish colonisation between the two experimental groups, siCTL or siSCN4B cells injected into the yolk sac, it is important to monitor the levels of SCN4B expression for the total duration of the experiment. As such, assessment of SCN4B expression by quantitative PCR has been performed in cells, 48h and 72h after siRNA transfection. Results are now given in the new Supplementary Figure 6a and indicate that SCN4B expression is down-regulated by $50.3 \pm 9.1\%$ and by $35.1 \pm 3.6\%$ in siSCN4B compared to siCTL, 72 h and 96 h after transfection. These results suggest that the down-regulation of SCN4B gene was still efficient at the end of the colonisation assay (72 h after transfection).

4. With regard to the electrophysiology and invasion data in Figure 3, a challenging observation to interpret is that when SCN4B is knocked down, there is no effect on persistent current, but the effect of TTX on invasion seems to be lost. Therefore, how/why does TTX have no effect in shSCN4B cells given that the channel is still functional and carrying a persistent current? The situation gets even more complex/interesting when considering the overexpression data in Figure 4, where beta-4 up-regulation seems to have the same effect as the RNAi, decoupling the effect of TTX on invasion, but at the same time, altering the current. I think these observations are important and they, together with possible explanations, should be discussed in more detail. Finally, what is the proposed mechanism for loss of transient current in the absence of beta-4 but no change in persistent current?

The referee is perfectly right, this was a bit confusing and difficult to interpret. Even though we always observed a small reduction of shSCN4B cell invasiveness in presence of 30 μ M TTX to inhibit Nav1.5 current, or when cells were transfected with a siRNA targeting SCN5A gene, we did not obtain a statistical difference.

We think that this was a statistical bias, because *i*) the number of samples was too small, and *ii*) we presented shSCN4B invasiveness as a ratio of shCTL cells. As a result, the dispersion of data was too wide to allow the identification of the effect of the persistent current attributed to Nav1.5 channels in shSCN4B cancer cell invasiveness.

This has been further studied, and by (solely) increasing the number of independent experiments (6 supplementary independent invasion experiments, in triplicate, were added), we can now record a significant reduction of shSCN4B invasiveness when inhibiting the activity (TTX) or expression (siSCN5A) of Nav1.5 channel. These results suggest that the persistent Nav1.5 current (found to be similar in the two cell types, whereas there was a reduction of the peak current in shSCN4B cells) participate equally in the invasiveness of aggressive cancer cells expressing SCN4B gene or hyper-aggressive cancer cells not expressing this gene. Please see the new figures 4a and 4b.

5. In Figure 2C, a loading control needs to be included (e.g. HSC70, as in Figure 2D). It would be helpful to also show the western indicating absence of beta-3 in MDA-MB-231 cells, together with positive and loading controls, since this result is mentioned in the text. The western in Supplementary Figure 7B also needs loading control, e.g. HSC70.

We thank the referee for these comments.

Concerning the loading controls (HSC70) of beta subunits in the figure 2C (new figure 3C), we just forgot to add them to the initial figure along with β 1, β 2 and β 4 protein expression. The mistake is repaired as they have been added.

MDA-MB-231 breast cancer cells have been already shown not to express the SCN3B gene, encoding for the β 3 protein. This has already been reported in Chioni *et al.*² and in Gillet *et al.*³. For this reason we did not perform supplementary western blotting experiments, but checked for the absence of mRNA expression using two different couples of PCR primers, and using Human Brain Total RNA extracts as a positive control.

The primers had the following sequences:

Couple of primers 1:

Forward 5'-GGCTGATCCCCCTAAGAGTC-3'

Reverse 5'-CGGCTTTTGAGACCTTTCTG-3'

Expected size of the amplicon: 147bp

Couple of primers 2:

Forward 5'-GAGGGCGGTAAAGATTTTCCT-3'

Reverse 5'-AGAGGCCAGAGTCGTTTTCAGA-3'

Expected size of the amplicon: 154bp

Results from these PCR experiments indicate that there was no cDNA amplification corresponding to the expression of *SCN3B* with these two couples of primers when using MDA-MB-231-Luc cDNA template, contrarily to that of the positive control condition using human brain extracts. Please see supplementary Figure 5a.

The two references cited above have been included in the text.

As demanded by the referee, a loading control has also been added to the western blot showing the overexpression of *SCN4B*/ β 4 protein in oe*SCN4B* cells, compared to CTL cells. Blots shown were obtained by exposing the same film at different time for these two conditions (1s for oe*SCN4B* and 30 s for CTL). Please see new Supplementary figure 8b.

6. In Figure 3N what is the beta-4 expression status across these different cell lines? It would be helpful to include a quick QPCR experiment here, or in the supplementary data, to show whether or not beta-4 expression is similar across this cell line panel.

We thank the referee for this comment. The expression level of *SCN4B*, assayed by RT-quantitative PCR experiments, across breast MDA-MB-468, non-small cell lung H460, non-small cell lung A549 and prostate PC3 cancer cells is now given in supplementary figure 5c. No statistical difference was found between these four cell lines.

Minor comments:

1. The expression data in Supplementary Fig 2 appear to be presented differently (log₂ median-centred intensity) than elsewhere in the manuscript (RKPD). Why is this? The different analytical approach for Supplementary Fig 2 needs to be added to the methods.

The referee is right. We should have explained this in the initial version of the manuscript. Expression data presented in Supplementary figure 2 (now in Supplementary Figure 3) come from other sets of experiments than those presented in Figure 2 (formerly in figure 1) and the method used was different.

Data presented here are coming from Affymetrix Human Genome U133 Plus 2.0 Array (and not from the total RNA sequencing such as presented in Figure 2, expressed in RPKM) coming from published studies^{4,5}. A sentence has been added to the legend of the Supplementary Figure 3.

2. The steady-state inactivation curve for sh*SCN4B* cells in Figure 3E looks to me to be biphasic. Have the authors tried fitting to a double-Boltzmann equation, (e.g. see Lopez-Santiago et al (2006) J Neurosci 26, 7984-7994)? A two-component fit might suggest the presence of two channel types, and/or incomplete inactivation.

We thank the referee for this interesting comment. It is true that the shape of the inactivation-voltage relationship for sh*SCN4B* cells looks somehow biphasic. However, we tried and compared fitting it with either single or double-Boltzmann equations, and the curve is better fitted with single one (with an adjusted R² of 0.99864).

3. For the immunohistochemical quantification in Fig 1: was the analysis performed blinded to tumour type/outcome data? The method of blinding (or lack of it) should be mentioned in the methods.

Immunohistochemical experiments on tissue microarrays, quantification and analyses of expression levels of $\beta 4$ in cancer (with the different grades) and some of normal tissues were performed from by our collaborators Carlos Martinez-Caceres (anatomopathologist), and Pablo Pelegrin (Immunologist, Group leader, Inflammation and Experimental Surgery Unit, Murcia's BioHealth Research Institute IMIB-Arrixaca, Clinical University Hospital Virgen de la Arrixaca, Murcia, Spain). Tissue microarrays and anti- $\beta 4$ antibodies were ordered (SR) then send to them for performing IHC and analyses that were performed blind, since they were not working in the field of sodium channels, did not know our initial project and preliminary results on $\beta 4$, and what we could expect for IHC staining.

Of course because C.M.-C. is anatomopathologist, he could recognize very easily normal epithelial tissues and the different grades of cancer in the TMA.

This is now mentioned in the methods.

4. In supplementary Figure 4D some other alpha subunits are downregulated at the mRNA level in shSCN4B cells. Why is this? This should be discussed.

The referee is right, it seems that the down-regulation of SCN4B gene in MDA-MB-231-Luc cancer cells also reduce the expression, at the mRNA level of *SCN1A*, *SCN4A* and *SCN5A* genes. In these cells, we³ and others⁶ have shown that sodium current is solely due to the activity of the Nav1.5 channel coming from *SCN5A* gene expression. The other gene isoforms are very weakly expressed at the mRNA level, but the presence of corresponding proteins has never been investigated. We must confess that we absolutely don't know the signalling involved in these gene down-regulations nor the biological relevance of these. Because this is out of the scope of this study, and even though this would be something interesting to study, we rather not comment too much about this in the present manuscript. We hope that the referee will understand our point of view.

Reviewer 2

The authors investigate the role of SCN4B, which encodes for beta4 protein originally described as an auxiliary subunit of NaV channels. They show that beta2 is expressed in epithelial tissues, but that expression disappears as tumors become more aggressive. Using in vitro assays, they show that knockdown beta4 leads altered cell morphology, and increases in 2D migration speed, invasion through matrigel, and RhoA activity; overexpression of the protein reduces invasion, with all effects being independent of NaV. The authors used xenograft models to examine these phenotypes in vivo, looking at primary tumor growth and metastasis. This is an interesting paper and, in principle, could be a nice contribution to the field. The authors should perform some additional cell biological experiments to support their hypothesis that SCN4B depletion causes a shift from mesenchymal to amoeboid motility/invasion, and to characterize the cell biological effects of SCN4B a bit further using standard assays.

We sincerely thank the reviewer for his/her constructive comments and advices he/ she formulated on our initial manuscript. We made our best efforts to answer the questions raised and have performed all requested experiments. We think that these comments helped us to improve the quality and to strengthen the relevance of our study.

Major comments:

-The 2D migration and matrigel transwell invasion assays are supportive of potential amoeboid like transitions in the SCN4B depleted cells, as are the data on increased RhoA activity and decreased matrix degradation/invasion, but these are not entirely satisfying given that true amoeboid migration cannot be assessed in 2D (or in transwell matrigel assays). More compelling data would come from performing 3D migration/invasion into collagen gels +/- GM6001 to determine whether SCN4B knockdown allows for protease independent migration.

We thank the referee for this comment and advice. We have performed and monitored shCTL and shSCN4B single cell invasion in 3D matrices composed of matrigel (2.7 mg/mL).

Invasion was monitored in time-lapse microscopy and analysed from pictures that were recorded every 30 min, starting 4 h after cell seeding, for a total duration of 48 h. These experiments were performed in the absence or presence of GM6001.

Analyses were performed from 2D images (X-Y axes) because we do not have the possibility to do it in 3D (X-Y-Z). Results have been included in the new figure 5e and 5f. They indicate that the loss of expression of SCN4B potentiates 3D invasiveness of breast cancer cells, and this induction is not completely abrogated by GM6001. These results further suggest that SCN4B-depleted cells display amoeboid transitions.

-The changes in protease activity assessed by the "focalized degradation index" are suggestive of changes in invadopodia formation/function. The authors should explore this using standard invadopodia assays and marker analysis.

The focalized degradation index represents a quantification of focalised areas of F-actin that merge with extracellular matrix degradation (proteolysis of the fluorescent DQ-gelatin). As such, this could be assimilated to a quantification of invadopodia degradative function⁷. We did not see any difference of degradative activity following this methodology between shCTL and shSCN4B cells. As a result, we believe that there is no difference in invadopodia function in these two cancer cell lines, and that they degrade the extracellular matrix similarly.

We agree with the reviewer that this methodology however does not allow the assessment of invadopodia formation and recycling.

Because the referee asked, we tried to co-localize proteins that are known to be enriched in invadopodia during their formation/maturation, with extracellular matrix degradation (DQ-gelatin fluorescence). We initially tried to label Tks5 as a marker of invadopodia, but only had a very diffuse signal in the perinuclear area. Therefore we labelled the phosphorylated form (Y421) of the actin nucleation-promoting factor cortactin, as a marker of invadopodia⁸, and proceeded as similarly indicated. These results are indicated in the new Supplementary Figure 5b). There was no difference in P-cortactin/DQ-gelatin degradation merging areas between shCTL and shSCN4B cells, further suggesting that, in these two cell types, invadopodia function is equivalent.

- SEM cannot be used to distinguish between filopodia and retraction fibers (4F), and the F-actin labeling in 4E does not show filopodia, which should be evident (perhaps there was insufficient exposure time or the focal plane with filopodia was missed). Filopodia should be quantified using actin labeling, preferably with filopodial markers (e.g. fascin, VASP), or by time-lapse imaging.

We thank the referee for this comment. To analyse the presence of filopodia, we initially tried to label Fascin 1, using the antibody sc-21743 from Santa-Cruz Biotechnology. However, the labelling was not good and even in confocal imaging we were only able to observe a punctate labelling in all the cytosol. Therefore, as suggested by the reviewer we performed F-actin labelling using phalloidin-AlexaFluor488 and used confocal imaging. These experiments allowed us to monitor the number, but also the length of filopodia in shCTL and shSCN4B cells. In agreement with results previously obtained in SEM, shSCN4B cells displayed less filopodia than shCTL cells (median were 8 vs. 24.5 filopodia/cells for shSCN4B and shCTL, respectively, $p < 0.001$, Fig. 6e-f). Furthermore, in shSCN4B cells, filopodia were of smaller length than in shCTL cells (median were 3.43 μm vs. 5.61 μm in shSCN4B and shCTL, respectively, $p < 0.001$, Fig. 6e-g). These analyses further suggest that the loss of SCN4B expression supports amoeboid transition.

- IHC quantification, in Fig 1, the authors quantify levels of beta4. Does the number of samples shown in 1a represent the number of fields? Or the number of patients? Was there one score per core, or multiple per core according to field of view? Please clarify in methods.

For IHC quantification performed from TMA, the number of samples is representative of the number of core, equivalent to the number of patients. There was only one score given per core. This is now clarified in the methods section.

- Expression of SCN4B is restricted to epithelial tissues: The authors write 'This suggests that SCN4B/ β 4 is normally expressed in normal epithelial cells (as confirmed by immunohistochemical analyses of normal oesophagus, (Suppl. Fig. 3)' The relevance of the staining of oesophagus tissue is not clear and not related to the rest of the study. It would be more interesting to perform co-stains for epithelial cell markers, or show staining of normal lung to show epithelial localization, or perhaps normal prostate, colon or prostate, as these data are referred to in Fig S2.

Yes, the referee is right, and to make this clearer we have performed new IHC experiments on normal breast and cancer breast biopsies coming from the University-Hospital of Tours (Tumour collection declared to the French Ministry of Research (N°DC2008-308). These have been performed by our anathomopathologist Prof. Gaelle Fromont (clinician at the University-Hospital of Tours) with the support of Dr Armelle Vigneux (acknowledged in the "Acknowledgments" section), and the results are presented in the new Figure 1. The immunolabelling of β 4 protein was very strong in epithelial cells of mammary acini, and was absent in non-epithelial cells of normal breast tissues (Fig 1a). By comparison, labelling of β 4 protein was significantly reduced in cancer cells (tumour area "T", indicated by the red arrow, Fig 1b). According to our anathomopathologist, the labelling in normal epithelial cells and structures is very clean and clear, and does not need co-staining with an epithelial cell marker.

- Expression of SCN4B in cell lines (Figure 2): In Fig 2a-c, authors show mRNA levels in multiple breast

cell lines. How does the protein expression compare? Are the differences seen at the mRNA level reflected in the protein?

SCN4B/ β 4 protein expression has been evaluated by western blotting experiments in the same cell lines (5 independent experiments) that were initially studied by RT-qPCR. The profiles of SCN4B/ β 4 protein expression (relative to HSC70) are very similar to those observed at the mRNA level in RT-qPCR. The non-cancer cell line MCF-10A express the SCN4B/ β 4 protein at significantly higher levels than the breast cancer cell lines. Please see new Figure 3b.

In fig 2D, does knockdown of one of the SCN genes affect expression (protein/mRNA) of the others? Please show these controls.

Similarly, in Fig3, does knockdown of SCN5A affect SCN4B levels?

In this revised version of the manuscript, Figure 2d became Figure 3e.

We thank the referee for this important question, referring to a potential compensation between subunits. As shown in the new Supplementary Figure 6d (former supplementary figure 4c in the initial version) knocking down the expression of *SCN4B* gene had no effect on the expression of the other *SCNx*B gene at the mRNA level. We now show that it does not affect either the protein level. Please see new Supplementary figure 6e.

Knocking-down the expression of *SCN1B*, *SCN2B*, or *SCN5A*, using specific siRNA sequences had no effect on the mRNA expression of one of the other *SCNx*B gene expression. This was assessed by RT-quantitative PCR. This is now indicated in the text.

- Increased association of RhoA with SCN4B (Fig 4m). How do these data fit in to the hypothesis proposed by the authors? Does SCN4B bind directly to Rho to keep it inactive or prevent its activation?

The referee is right, this is something intriguing. Yes, we believe that the presence of SCN4B/ β 4 protein at the plasma membrane sequester RhoA and keep it inactive. Of course this will have to be further studied, since we have no real proof of this at the moment.

-overexpression experiments- generation of overexpressing cell lines: FigS7b- what are all the bands in the overexpression? Does overexpression lead to different isoforms or PTMs that affect MW? Why do the authors chose to to overexpress SCN4B by 40,000K fold overexpression at the mRNA level? This is likely not physiologically relevant.

Corresponding western blots are now showed in the new supplementary figure 8b (previously in Suppl. Fig. 7b). We performed new western blotting experiments and blots are better than in the initial version (which were probably overexposed). We do not think that the overexpression of the *SCN4B* gene leads to different isoforms, and we can see here a unique band at the expected molecular weight). Also, as demanded by the referee 1, we added a loading control (HSC70).

It is true that the level of *SCN4B* expression is not “physiologically relevant” (even if we are working on cancer cell lines that, by definition, are not physiological), however, we did not choose to overexpress to such a high level. Of course, the best would be to express *SCN4B* in cancer cells in similar levels that what is recorded in normal epithelial cells. Unfortunately, it is not that easy to have a good expression vector with a promoter strong enough to ensure the efficient expression of the transgene. Also, it was important to us to generate a stable cell line (for injection into immunodepressed mice), in which all individual cells overexpressed *SCN4B* gene, to be used in single cell experiments (such as patch clamp recordings).

Fig5H- authors should show representative image of oe cells
Representative images of oeCTL and oeSCN4B cells are now shown in Supplementary Figure 9a.

Fig5L-Orescue experiments: what is the level of expression for these constructs? 40,000 X? or back to normal levels.

All the constructs were inserted into the pSecTag2 hygro B expression vector, and therefore under the dependence of the same promoter. They are all expressed at similar levels in sh*SCN4B* cells that have been selected not to express *SCN4B* gene. It is therefore difficult to quantify the real expression level of the protein variant. To assess this, plus the subcellular localization of variants, we performed immunostaining. Images show similar expression levels of fragments in transfected sh*SCN4B* cells, and suggested that they were all addressed at the plasma membrane (please see Supplementary figure 9b).

Please adjust language for Figure 5M in the results: the full length only rescues 50% phenotype, but not all of it.

Yes, the referee is right and we have corrected this in the text:

“Interestingly, the Δ C-ter variant, which possessed the extracellular domain, was completely ineffective, whereas the Δ N-ter variant reduced cell invasiveness to the same extent as the full-length protein (Fig. 7m). The invasiveness of sh*SCN4B* cells overexpressing the Δ N-ter variant or the full-length protein was still superior to the one of shCTL cells by about 50%.”

- *in vivo* experiments: It is unclear why the authors chose to not show data from a control cell line in any of these experiments. Does overexpression or knockdown of *SCN4B* induces changes in either direction that are significantly different from control cells? It is hard to evaluate the significance of these findings without the right controls.

We understand the question of the referee. We were probably not clear enough in the original submission. We initially formulated the hypothesis that the *SCN4B* gene could represent a potential metastases suppressor. In the *in vivo* experiments performed in immunodepressed mice, we chose to test our hypothesis that the *SCN4B* gene and therefore its expression product, the β 4 protein, could prevent the development of metastases. We know that parental MDA-MB-231 cells are aggressive breast cancer cells that form metastases in nude mice, and they could not represent good control cells since they express the *SCN4B* gene at a low level compared to non-cancer cells. Therefore we chose to compare metastases appearance in mice injected with breast cancer cells expressing (to a sufficient level, close enough to what is found in non-cancer cells) or absolutely not expressing (such as in high grade tumours) the *SCN4B* gene. That is the reason why we compared oe*SCN4B* cells to sh*SCN4B* cells in our two *in vivo* models. These two cell lines were just models to test the effect of *SCN4B* expression (or not) on tumour growth and metastases development. We have clarified this in the text.

Ideally, the best would have been to compare cells that express *SCN4B* at levels that are similar to non-cancer cells, to cells not expressing *SCN4B*. But obviously non-cancer cells are non-tumorigenic. For this revision we tried to knock down the expression of *SCN4B* in the non-cancer mammary cell line MCF-10A to assess whether they become invasive *in vitro* and tumorigenic *in vivo*. We successfully obtained cells that were not expressing the *SCN4B* gene, however they were no longer proliferating. We don't know whether this was due to a real biological effect following *SCN4B* knock down, or whether this was an artefact. We could not reproduce this experiment during the time that was given to us for this revision. We however decided that we should further explore this in the future. Emeline Bon, first author of the manuscript finished her contract at the end of July 2016 and we have no possibility to run other *in vivo* experiments at the moment.

We hope that the referee will understand our position.

Fig6e- how do the authors explain the differences in primary tumor size? Given the differences in primary tumor size, the authors should quantify metastatic index (numbers of mets relative to tumor size) to measure metastatic potential.

Yes, this is true that the loss of *SCN4B* expression had important consequences in promoting primary tumour growth. He showed here that, in *in vitro* experiments, the loss of *SCN4B* expression did not affect cell survival and proliferation (Suppl. Fig. 6c). One possible explanation for the effect observed *in vivo* is that, in mice injected with sh*SCN4B* cells, the increase of local invasion allows for a better tumour growth.

Importantly, mice implanted with sh*SCN4B* cells had more metastatic foci (5/8 mice) compared to those implanted with oe*SCN4B* cells (only 1/8 mouse). We cannot run statistical tests to compare the size of

metastases in shSCN4B and oeSCN4B groups (only one metastasis in oeSCN4B), however, the BLI signal of the unique metastasis identified in the oeSCN4B group was in the same range than the BLI signals recorded for the five metastases identified in the shSCN4B group. As suggested by the referee we calculated a “Metastatic index” as being the ex vivo BLI of metastases relative to the ex vivo BLI of the primary tumour. The metastatic index is higher in the shSCN4B group (supplementary Figure 10e)

- Scale bars: Many of the images are missing scale bars, please add these. Please add scale bar to images in 1b and d., also S2D, 2E

We apologize for this. We have added scale bars to all images in which they were initially missing.

Minor comments:

p4: 'The levels of $\beta 4$ expression were similar in mammary hyperplasia and dysplasia. 'Clarify-similar normal breast vs hyperplasia or dysplasia? Or similar between hyperplasia and dysplasia?

The referee is right, the sentence was not clear. This has been corrected: “The levels of $\beta 4$ expression were similar in mammary hyperplasia and dysplasia, compared to normal mammary tissues, but were remarkably reduced in biopsies of mammary carcinomas.”

p.5 'Importantly, SCN1B and SCN4B genes appeared to be the two most highly expressed genes in non-cancer tissues (Suppl. Fig. 1g), and SCN4B the most significantly down-regulated gene in breast cancer tissues (Suppl. Fig. 1h).' This is confusing, sounds like Scn4B is the two most highly expressed gene out of all the genes. Is this what the authors meant? Or is it within this family of genes?

We apologize for this sentence that was not clear. We have changed as follows:

“Importantly, *SCN1B* and *SCN4B* genes appeared to be the two most highly expressed *SCNxB* genes in non-cancer tissues (Suppl. Fig. 2g). Furthermore, *SCN4B* seemed to be the most significantly down-regulated *SCNxB* gene in breast cancer tissues compared to non-cancer tissues (Suppl. Fig. 2h).”

Figure 4m- shctrl and shSCN4B representative images and quantification should all be next to each other in the same panel for better comparison

This is now done in the new figure 6l.

Figure 3D- Does overexpression of SCN5a in shSCN4B rescue the Na phenotype?

This is a very good question. We tried this, but failed so far to overexpressed *SCN5A* gene, which is very big, in these cells.

It seems from this study that the loss of *SCN4B* expression reduces the expression of *SCN5A* at the mRNA level (qPCR experiments) and the reduction of maximal sodium current in electrophysiology experiments is also suggestive for a reduction of protein expression and/or addressing to the plasma membrane. However, we demonstrated that loss of *SCN4B* expression also increase the activity of (remaining) $\text{Na}_v1.5$ channels through an increased proportion of persistent current (please see the intersection between activation- and availability-voltage relationships in Fig 4e). As a result the absolute level of persistent sodium current is absolutely the same (Fig 4f-g). Since this current was associated to the mesenchymal invasion of breast cancer cells^{7,9}, we could anticipate that the overexpression of *SCN5A* in *shSCN4B* cells would increase the ECM degradative potency of cells, but only if the expressed channels display a window of voltage.

References cited in this letter

- 1 Wong, H. K. *et al.* beta Subunits of voltage-gated sodium channels are novel substrates of beta-site amyloid precursor protein-cleaving enzyme (BACE1) and gamma-secretase. *J Biol Chem* **280**, 23009-23017 (2005).
- 2 Chioni, A. M., Brackenbury, W. J., Calhoun, J. D., Isom, L. L. & Djamgoz, M. B. A novel adhesion molecule in human breast cancer cells: voltage-gated Na⁺ channel beta1 subunit. *Int J Biochem Cell Biol* **41**, 1216-1227, doi:10.1016/j.biocel.2008.11.001 (2009).
- 3 Gillet, L. *et al.* Voltage-gated Sodium Channel Activity Promotes Cysteine Cathepsin-dependent Invasiveness and Colony Growth of Human Cancer Cells. *J Biol Chem* **284**, 8680-8691 (2009).
- 4 Okayama, H. *et al.* Identification of genes upregulated in ALK-positive and EGFR/KRAS/ALK-negative lung adenocarcinomas. *Cancer Res* **72**, 100-111, doi:10.1158/0008-5472.CAN-11-1403 (2012).
- 5 Hou, J. *et al.* Gene expression-based classification of non-small cell lung carcinomas and survival prediction. *PLoS One* **5**, e10312, doi:10.1371/journal.pone.0010312 (2010).
- 6 Fraser, S. P. *et al.* Voltage-gated sodium channel expression and potentiation of human breast cancer metastasis. *Clin Cancer Res* **11**, 5381-5389 (2005).
- 7 Brisson, L. *et al.* NaV1.5 Na⁺ channels allosterically regulate the NHE-1 exchanger and promote the activity of breast cancer cell invadopodia. *J Cell Sci* **126**, 4835-4842 (2013).
- 8 Saltel, F. *et al.* Invadosomes: intriguing structures with promise. *Eur J Cell Biol* **90**, 100-107 (2011).
- 9 Driffort, V. *et al.* Ranolazine inhibits NaV1.5-mediated breast cancer cell invasiveness and lung colonization. *Mol Cancer* **13**, 264, doi:10.1186/1476-4598-13-264 (2014).

Reviewer #1 (Remarks to the Author)

The authors appear to have addressed the comments/suggestions of both reviewers very well. I have only one minor comment: in suppl. fig. 8b, on the copy I have open on my computer it looks like the two lanes for the SCN4B western (CTL and oe) have been cropped and joined together, possibly from different westerns. I am not sure what this journal's policy is regarding cropping and joining lanes together on western images, but a vertical dotted line between cropped images to indicate this fact, plus mention in the legend, would be helpful, or better still, the western should be repeated with a better image of adjacent lanes that are uncropped/not artificially joined. Other than that, the manuscript seems fine.

Reviewer #2 (Remarks to the Author)

The revised manuscript has been improved and is now suitable for publication in Nat Comm

Reviewer 1

The authors appear to have addressed the comments/suggestions of both reviewers very well.

I have only one minor comment: in suppl. fig. 8b, on the copy I have open on my computer it looks like the two lanes for the SCN4B western (CTL and oe) have been cropped and joined together, possibly from different westerns. I am not sure what this journal's policy is regarding cropping and joining lanes together on western images, but a vertical dotted line between cropped images to indicate this fact, plus mention in the legend, would be helpful, or better still, the western should be repeated with a better image of adjacent lanes that are uncropped/not artificially joined. Other than that, the manuscript seems fine.

Yes, this figure comes from the same western blotting experiments but the pictures have been taken with two exposure times so that we can see proteins in both conditions.

As suggested by the reviewer, we now have added a vertical dotted line between the two exposure times of the same WB experiment, as indicated in the figure legend.

.